# Performance Evaluation of THz Atmospheric Limb Sounder (TALIS) of China

Wenyu Wang[1,2], Zhenzhan Wang[1], Yongqiang Duan[1,2]

[1]Key Laboratory of Microwave Remote Sensing, National Space Science Center, Chinese Academy of Sciences, Beijing, China

[2]University of Chinese Academy of Sciences, Beijing, China

*Correspondence to*: Zhenzhan Wang (wangzhenzhan@mirslab.cn)

**Abstract.** THz Atmospheric Limb Sounder (TALIS) is a microwave limb sounder being developed for atmospheric vertically resolved profile observations by the National Space Science Center, Chinese Academy of Sciences (NSSC, CAS). It is designed to measure the temperature and chemical species such as $O_3$, $HCl$, $ClO$, $N_2O$, $NO$, $NO_2$, $HOCl$, $H_2O$, $HNO_3$, $HCN$, $CO$, $SO_2$, $BrO$, $HO_2$, $H_2CO$, $CH_3Cl$, $CH_3OH$, and $CH_3CN$ with high vertical resolution from about 10-100 km to improve our comprehension of atmospheric chemistry and dynamics, and to monitor the man-made pollution in the atmosphere. Four heterodyne radiometers including several FFT spectrometers of 2 GHz bandwidth with 2 MHz resolution are employed to obtain the atmospheric thermal emission in broad spectral regions centred near 118, 190, 240, and 643 GHz. A theoretical simulation is performed to estimate the retrieval precision of the main targets and to compare them with that of Aura MLS standard spectrometers. Single scan measurement and averaged measurement are considered in simulation, respectively. Temperature profile can be obtained with the precision of < 2K for a single scan from 10 to 60 km by using 118 GHz radiometer, and the 240 and 643 GHz radiometer can provide temperature information in the upper troposphere. Chemical species such as $H_2O$, $O_3$, $HCl$ show relatively high single scan retrieval precision of < 20% over most of the useful range and $ClO$, $N_2O$, $HNO_3$ can be retrieved with a precision < 50%. The other species should be retrieved by using averaged measurements because of the weak intensity and/or low abundance.

## 1 Introduction

Better precision observation of Earth's atmosphere is essential to the numerical weather prediction and climate change studies. Satellites can provide daily global coverage of the atmosphere. Instruments such as nadir microwave sounder and infrared sounder have been applied to measure the atmospheric temperature and humidity but with the poor vertical resolution and altitude range (Swadley et al., 2008). Limb sounders can not only provide the temperature profile with better vertical resolution but gather information on chemical composition in a wide altitude range. In the terahertz domain, the measurement performances are independent of the day-night cycle. Microwave limb sounding is a particularly useful

technique in detecting stratospheric and mesospheric temperature and chemistry, and also has large potential for global wind measurement in the middle and upper atmosphere (Wu et al., 2008; Baron et al., 2013).

A few instruments have been launched in recent twenty years, their observation data have offered a better understanding of the physical and chemical processes in the Earth's atmosphere. The first instrument applying the microwave limb sounding technique from space was the Microwave Limb Sounder (MLS) onboard the Upper Atmosphere Research Satellite (UARS) launched in 1991. The sounder offered unique information of temperature/pressure, $O_3$, $H_2O$, ClO, and additional data products including $SO_2$, $HNO_3$, and $CH_3CN$ (Waters et al., 1993; Barath et al., 1993; Waters et al., 1999). The Sub-Millimetre Radiometer (SMR) onboard the Odin satellite launched in February 2001 was the first radiometer to employ sub-millimetre in limb sounding. Various target species such as $O_3$, ClO, $N_2O$, $HNO_3$, $H_2O$, CO, NO, as well as isotopes of $H_2O$, $O_3$, and ice cloud have been detected (Murtagh et al., 2002; Urban et al., 2005; Eriksson et al., 2007). Aura MLS, the follow-on of UARS MLS onboard the Aura satellite launched in July 2004 gave successful observations of OH, $HO_2$, $H_2O$, $O_3$, HCl, ClO, HOCl, BrO, $HNO_3$, $N_2O$, CO, HCN, $CH_3CN$, $SO_2$, ice cloud, and wind (Waters et al., 2004; Waters et al., 2006; Wu et al., 2008; Livesey et al., 2013). The Superconducting Submillimeter-wave Limb-Emission Sounder (SMILES) onboard the Japanese Experiment Module (JEM) of the International Space Station (ISS) launched in September 2009 (Kikuchi et al., 2010). SMILES was equipped with 4K cooled Superconductor–Insulator–Superconductor (SIS) mixers to reduce the system noise temperature so that the sensitivity of the SMILES was higher than that of other similar sensors such as MLS and SMR (Takahashi et al., 2010; Baron et al., 2011). Currently, several new instruments are being developed. Stratospheric Inferred Winds (SIW) is a Swedish mini sub-millimetre limb sounder for measuring wind, temperature, and molecules in the stratosphere. It can provide horizontal wind vectors within 30–90 km, as well as the profiles of temperature, $O_3$, $H_2O$ and other trace chemical species (Baron et al., 2018). SIW is designed for small satellites and will be launched as early as 2020–2022. In addition, the follow-on of SMILES, SMILES-2, is being studied for measuring the whole vertical range of 15−180 km with low noise (Ochiai et al., 2017).

THz Atmospheric Limb Sounder (TALIS) is the pre-research project of civil aerospace technology proposed by China National Space Administration (CNSA). TALIS is being designed at National Space Science Center, the Chinese Academy of Sciences (NSSC, CAS) for good precision measurement of atmospheric temperature and key chemical species. It has four microwave radiometers in the frequency bands of 118, 190, 240, and 643 GHz which are similar to Aura MLS. TALIS mission objectives are to provide the information for research on the dynamics and the chemistry of the middle and upper atmosphere by measuring the volume mixing ratio (VMR) profile of the chemical species and other atmospheric condition such as cirrus with much finer spectral resolution. The pre-research will be completed in 2020 and a prototype will be tested. The satellite mission equipped with TALIS will be proposed around 2021.

In this paper, we present a simulation study on precision estimates for the geophysical parameters measured by TALIS. The outline of the present study is as follows: Section 2 describes the instrument characteristics and spectral bands. The

retrieval method and the simulation result are discussed in Sects. 3 and 4, respectively. The final section gives a conclusion about the performance and future works.

## 2 Instrument overview

### 2.1 Instrument characteristics

The TALIS payload (Fig. 1) and its proposed scan characteristics are summarized in Table 1. The instrument will be set at a sun-synchronous orbit at a normal altitude of 600 km. The offset parabolic antenna is made of a single reflector with 1.6 m projective aperture and four independent feeds. The layout of four discrete feeds is shown in Fig. 2. Compared with the quasi-optical separation layout (such as MLS), this strategy is easier and has better observation precision since it needs fewer reflectors. But it will lead to a vertical observed difference of about 20 km between 118, 190, and 643 GHz and horizontal

displacement of 240 GHz. The widths of the field of view (FOV) at the tangent point are about 5.5, 3.8, 3.3, and 0.96 km at 118, 190, 240, 643 GHz, respectively. The two-point calibration method is adopted by TALIS, and two calibration targets are set at the  end of the arm. The extra target can be used to improve the calibration precision and evaluate the antenna effect and nonlinearity. At the beginning of the scan, TALIS will view the hot target (ambient temperature) and the extra target (lower temperature) in 3 s, and then it will scan the limb from 0 to 100 km vertically and obtain the spectra every 1 km

with an integration time of 0.1 s, finally, it views the cold space at 200 km in 5 s. The process of retrace is the same (also record data) and giving a total period (scan and retrace) of about 36 s.

TALIS has four radiometers which cover the significant thermal emission spectra in the 118, 190, 240, and 643 GHz regions (see Table 2). Single-sideband (SSB) can keep the complete spectral lines while double-sideband (DSB) can cover more spectral lines because of the image band. Thus, all the radiometers of TALIS will operate in the double-sideband mode

except the 118 GHz radiometer. Eleven FFT spectrometers of 2 GHz bandwidth with 2 MHz resolution will be used in TALIS. The bands and system noise temperature for each radiometer are shown in Table 2.

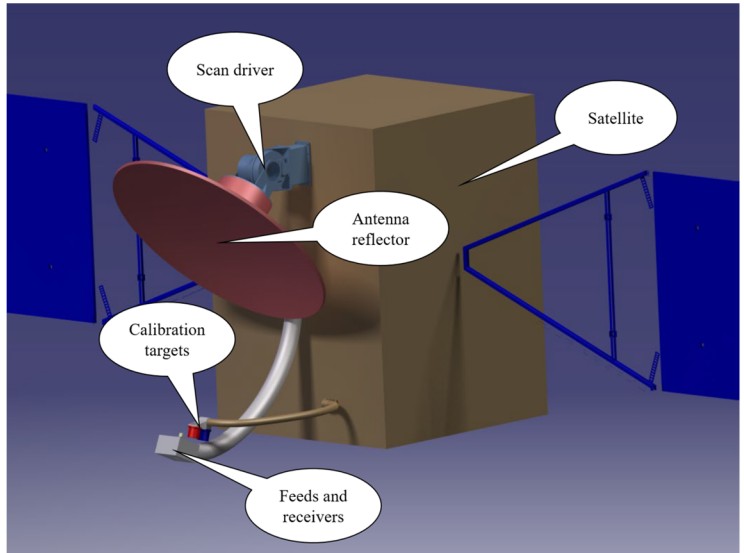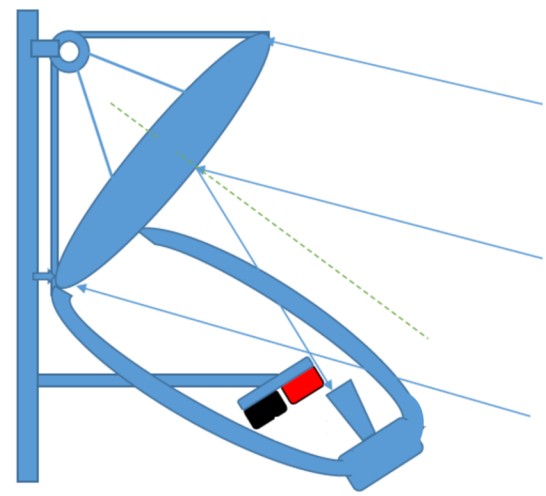

**Figure 1. The schematic diagram of TALIS payload. The reflector, feeds, and receivers are formed into a whole. The scan driver controls the scan angle. The calibration system is fixed in the satellite. At the beginning, the feeds are covered by calibration targets. Then it will scan the limb. When the system rotates to the top, it will view the cold space.**

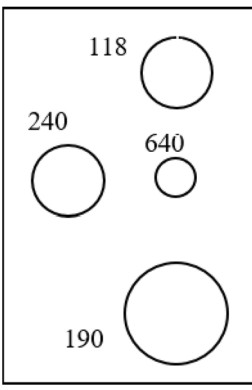

**Figure 2. The layout of the four antenna feeds. There exists a vertical observed difference of about 20 km between 118, 190, and 643 GHz and horizontal displacement between 240 GHz and other three radiometers.**

**Table 1. Characteristics of the TALIS payload**

| Satellite altitude | 600 km |
|---|---|
| Vertical scanaltitude | 0–100 km |
| LOS nadir angle | 66.07–68.17° (2.1°) |
| Scan velocity | 0.21°s$^{-1}$ (36 s scan$^{-1}$) |
| Spectrum integration time | 0.1 s (1 km ) |
| Antenna size | 1.6 m |
| Antenna vertical FOV | 5.5, 3.8, 3.3, 0.96 km |
| Spectrometer Bandwidth | 2 GHz |
| Spectrometer resolution | 2 MHz |

| LO frequency | | | | 120, 190.1, 239.66, 642.87 GHz | | | |
|---|---|---|---|---|---|---|---|

**Table 2. Spectral bands and Tsys of TALIS**

| Radiometer | TALIS (GHz) | Tsys* for TALIS | Tsys for MLS | Radiometer | TALIS (GHz) | Tsys for TALIS | Tsys for MLS |
|---|---|---|---|---|---|---|---|
| 118 GHz | 117.75–119.75 | 1000 K | 1200 K | 240 GHz | 229.66–231.66 247.66–249.66 | 1000 K | 1200–1600 K |
| 190 GHz | 175.5–177.5 202.7–204.7 | 1000 K | 900–1100 K | | 232.16–234.16 245.16–247.16 | | |
| | 178.9–180.9 199.3–201.3 | | | | 234.66–236.66 242.66–244.66 | | |
| | 183.0–185.0 195.2–197.2 | | | 643 GHz | 624.47–626.47 659.27–661.27 | 2300 K | 4000–4400 K |
| | | | | | 627.37–629.37 656.37–658.37 | | |
| | | | | | 632.37–634.37 651.37–653.37 | | |
| | | | | | 634.87–636.87 648.87–650.87 | | |

\* This is a single-sideband value for 118 GHz radiometer, and double-sideband value for other radiometers.

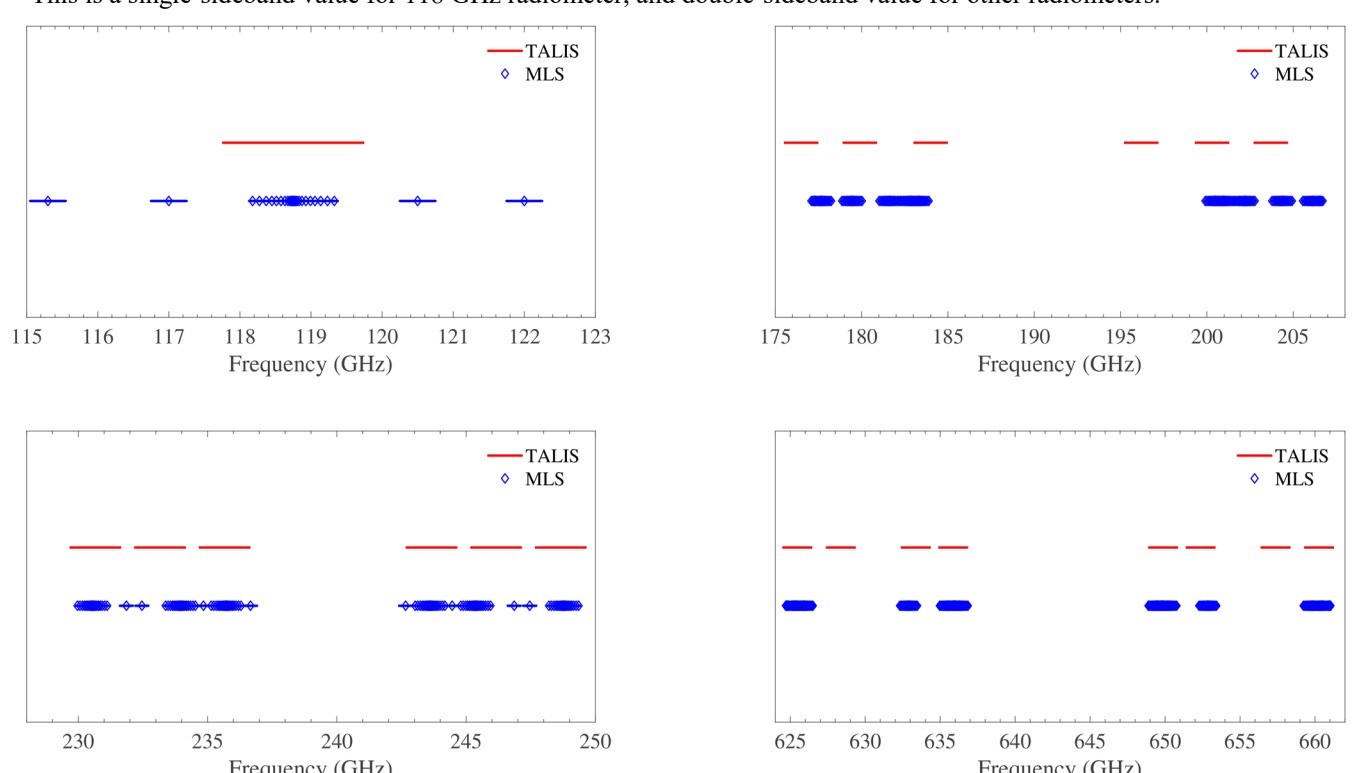

**Figure 3. Spectral bands of Aura MLS and TALIS radiometers. The diamond represents the line centers, and the solidline means**
5  **the bandwidth of MLS.**

## 2.2 Spectral bands

The spectral bands of TALIS are selected with the following criteria: (1) maximizing the number of species which will exert a strong influence on atmospheric chemistry and dynamics, (2) necessary space between the passband, (3) trade-off between realizable bandwidth and resolution. TALIS covers most spectral bands of Aura MLS and extends them (see Fig 3), but lack the 2.4 THz band. The broader bandwidth and the finer resolution of TALIS can provide better retrieval precision and effective altitude range compared with Aura MLS. More chemical species can be measured by TALIS, such as $NO_2$, NO, and $SO_2$ (normal concentration).

The 118 GHz radiometer, covering the strong $O_2$ line at 118.75 GHz, is used to measure the atmospheric temperature and tangent pressure. Since there are few meteorological data set about the temperature above the middle atmosphere with good vertical resolution, it is necessary to measure the temperature profile with wide altitude range, good vertical resolution, and good precision. In addition, the Zeeman effect will affect the $O_2$ line, the influence should be studied (Schwartz et al, 2006). Other information such as ice cloud can be treated as additional measurement. Figure 4 gives an overview of the 118 GHz spectral band.

The 190 GHz radiometer is mainly designed to cover the 183.31 GHz $H_2O$ line. Monitoring water vapour is important for understanding the mechanisms that humidity feedback on climate, and is essential for improving the accuracy of the weather forecast. Other chemical species such as $N_2O$, ClO, $O_3$, and HCN are also included in 190 GHz bands (see Fig. 5).

The main objective of the 240 GHz radiometer is to measure the CO at 230.54 GHz and the strong $O_3$ lines in a wide spectral band where upper tropospheric $O_3$ can be obtained with good precision because of the weak water vapour continuum absorption. In addition, the 233.95 GHz $O_2$ line will be used to measure temperature and tangent pressure together with 118.75 GHz line. $SO_2$ is an important pollutant in the Earth' atmosphere and will give rise to acid rain. There is no obvious $SO_2$ emission with the standard profile present in the passband of 240 GHz radiometer. The only $SO_2$ which is observable by MLS comes from volcanic eruptions. MLS demonstrated that $SO_2$ can be measured by 190 GHz, 240GHz, and 640 GHz radiometer, but only 240 GHz $SO_2$ product is recommended for general use (Pumphrey et al., 2015). The wide and strong lines of $HNO_3$ can be used to retrieve profile well. $NO_2$ is a unique species not covered by Aura MLS, and TALIS's wider bandwidth and finer resolution have the potential ability to measure it. The spectra of 240 GHz radiometer are depicted in Fig. 6.

The 643 GHz radiometer is designed to cover as many spectral lines as possible, thus about 17 species are included. The spectral lines covering $O_3$, HCl, ClO, $N_2O$, $O_2$, and $H_2O$ are clearly visible (Fig. 7), and other lines which are relatively weak such as NO, $HNO_3$, CO, $SO_2$, BrO, $HO_2$, $H_2CO$, HOCl, and $CH_3Cl$, can also be used. The $O_2$ line at 627.75 GHz and the $H_2O$ line at 657.9 GHz have the potential to be used as supplements to 118 and 190 GHz radiometers. $O_3$ is the major species in the stratosphere and mesosphere, which is quite important in atmospheric radiation transfer. Using the high sensitivity lines in the 643 GHz bands, one can measure $O_3$ with good precision (Takahashi et al., 2011; Kasai et al., 2013). The only lines of HCl below 1 THz are in the 625 GHz frequency band, thus HCl can be measured by 643 GHz radiometer (Lary and

Aulov, 2008). ClO is a key catalyst for ozone loss and the 649.45 GHz line is suitable for ClO observation with good precision (Santee et al., 2008; Sato et al., 2012). The HOCl, which will affect stratospheric chlorine budget, has distinct lines above 600 GHz, and the 635.87 GHz line has been pointed out to be the best line for observation (Urban, 2003). Both 649.701 and 660.486 GHz lines can be used to measure the hydroperoxyl radical $HO_2$, which will contribute to the catalytic

ozone chemistry in the upper stratosphere and mesosphere (Millán et al., 2015). Since ClO, $HO_2$, and HOCl all can be measured, the reaction rate of ClO and $HO_2$ to form HOCl in the atmosphere can be determined (Johnson et al., 1995). $N_2O$ can be measured at 652.834 GHz, which has been validated by MLS (Lambert et al., 2007). NO has two weak signals at 651.45 and 651.75 GHz which can be used to measure the abundance. $HNO_3$ can be measured using 650 GHz bands. Measuring these nitrogen species will help researchers to understand the chemistry and dynamics of the atmosphere better.

The BrO, which plays an important role in the depletion of ozone, can be measured using 624.768 and 650.179 GHz lines. Because of the low abundance of BrO, measurements must be significantly averaged in order to get reliable results (Millán et al., 2012). CO and $H_2CO$ are the major species in the $CH_4$ oxidation to $CO_2$ and $H_2O$ in the stratosphere and mesosphere (Suzuki et al., 2015). The major spectral line of CO used by MLS is at 230.538 GHz, however, the 661.07 GHz line can also provide information (Livesey et al., 2008). $H_2CO$ has a line at 656.45 GHz, but the signal is very weak. The $SO_2$ lines in the

660 GHz band have the potential to detect the background levels of $SO_2$. $CH_3Cl$ can be measured in the 649 GHz band near the line of ClO. MLS measured $CH_3OH$ and $CH_3CN$ in the troposphere and lower stratosphere by 625 GHz spectrometer (Pumphrey et al., 2011).

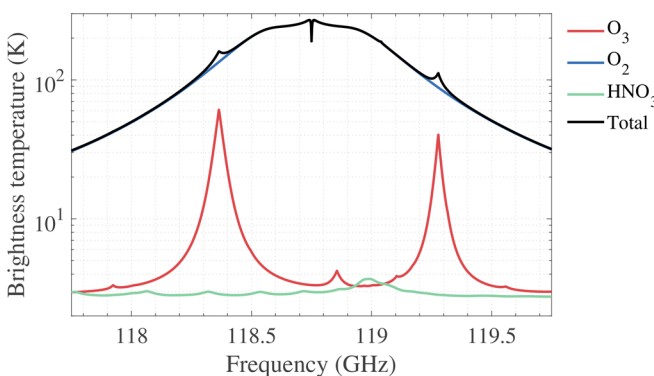

**Figure 4. Contributions of the main target chemical species to the 118 GHz spectrum. The brightness temperature is measured**

**from single sideband radiometer. The tangent height is 30 km.**

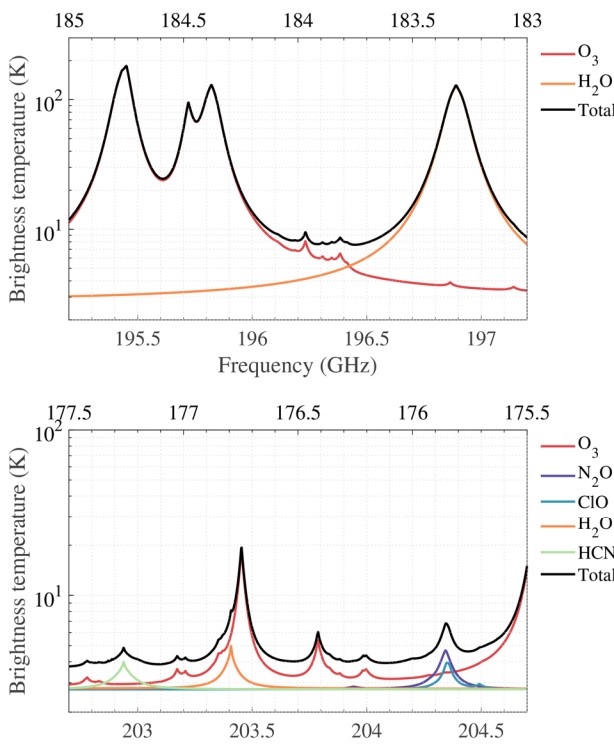

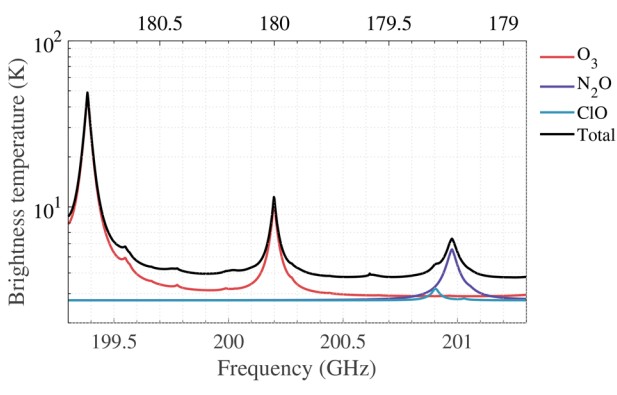

**Figure 5. Contributions of the main target chemical species to the 190 GHz spectra. The brightness temperature is measured from double sideband radiometer. The tangent height is 30 km. The top axis represents the lower sideband frequencies and the bottom axis represents the upper sideband frequencies. Each panel represents a single spectrometer.**

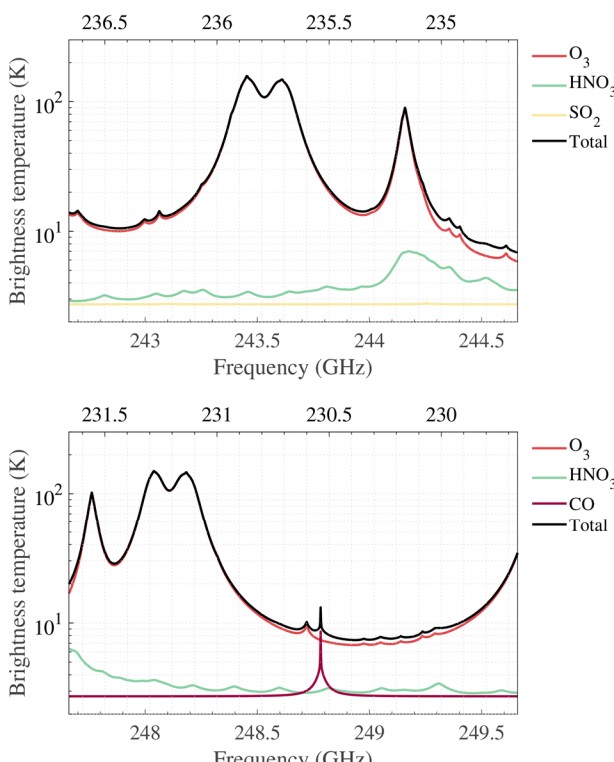
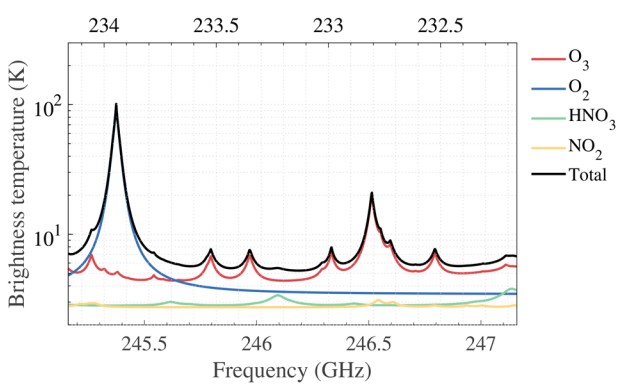

Figure 6. Contributions of the main target chemical species to the 240 GHz spectra. The brightness temperature is measured from double sideband radiometer. The tangent height is 30 km. The top axis represents the lower sideband frequencies and the bottom axis represents the upper sideband frequencies. Each panel represents a single spectrometer.

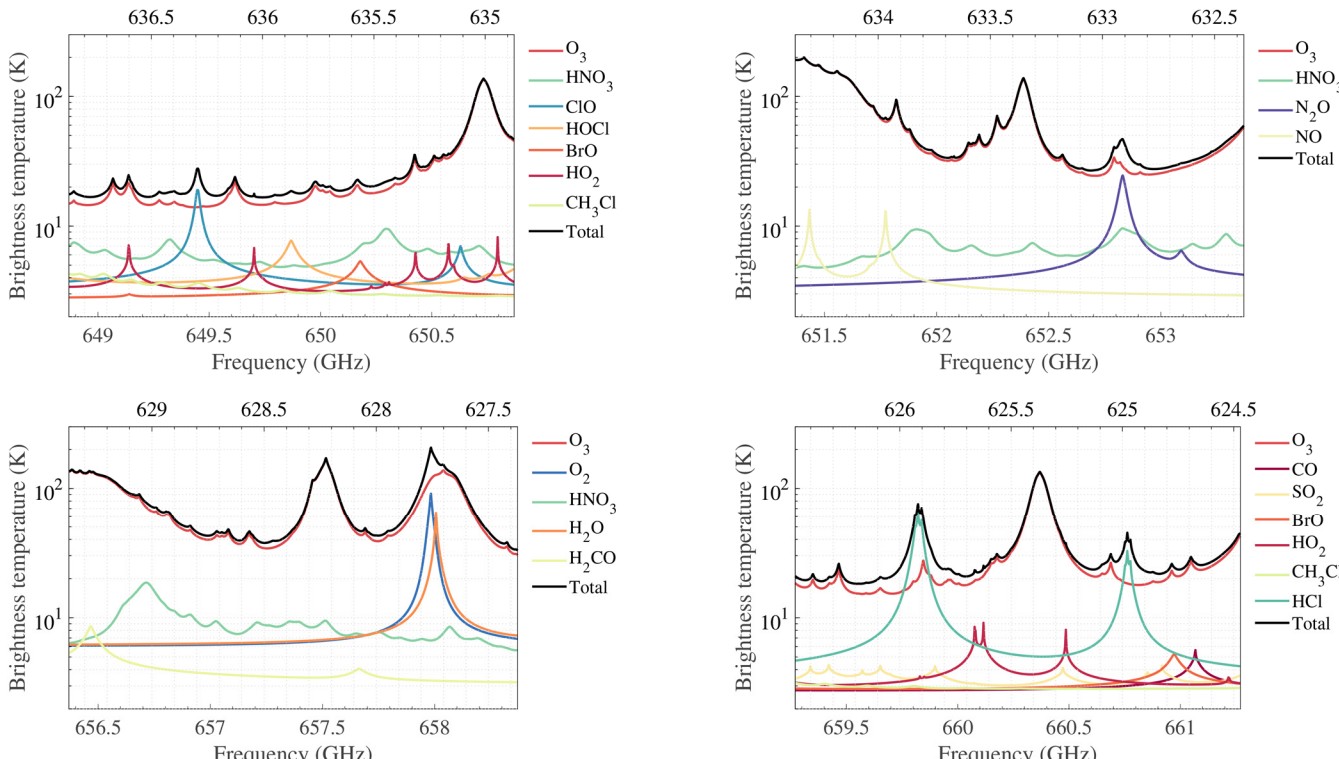

**Figure 7. Contributions of the main target chemical species to the 643 GHz spectra. The brightness temperature is measured from double sideband radiometer. The tangent height is 30 km. The top axis represents the lower sideband frequencies and the bottom axis represents the upper sideband frequencies. Each panel represents a single spectrometer.**

## 3 Retrieval methodology

### 3.1 Forward model

The retrieval of data measured by microwave limb sounder requires the accurate simulation of the observed thermal emission spectra. The forward model is a mathematical tool used to describe the radiative transfer, spectroscopy, and instrumental characteristics. The output of the forward model is the convolution of atmospheric radiation and instrument response.

Radiative transfer describes the emission, propagation, scattering, and absorption of electromagnetic radiation (Mätzler., 2006). Scattering can usually be neglected above the upper troposphere as the atmosphere is largely cloud-free at these altitudes, and such clouds as there are (e.g., Polar Stratospheric Clouds) have particle sizes shorter than the TALIS observation wavelengths. In this way and assuming Local Thermodynamic Equilibrium (LTE), the formal solution of the radiative transfer equation is defined by

$$I_v(S_2) = I_v(S_1)e^{-\tau_v(S_1,S_2)} + \int_{S_1}^{S_2} \alpha_v(s)B_v(T)e^{-\tau_v(S,S_2)}ds, \tag{1}$$

where $I_v$ is the radiance at frequency $v$ reaching the sensor, $\alpha$ is the absorption coefficient and $\tau$ is the opacity or optical thickness. $B_v$ stands for the atmospheric emission which is given by Planck function describing the radiation of a black-body at temperature $T$ and frequency $v$ per unit solid angle, unit frequency interval and unit emitting surface (Urban et al., 2004):

$$B_v(T) = \frac{2hv^3}{c^2} \frac{1}{e^{hv/k_BT}-1}, \tag{2}$$

where $h$ is the Planck constant, $c$ is the speed of light, $k_B$ denotes Boltzmann constant.

Spectroscopy models and databases allow us to compute the absorption coefficient which requires pressure, temperature, and the species concentrations along the line of sight. The basic expression can be written as:

$$\alpha(v) = nS(T)F(v), \tag{3}$$

where $S$ is called the line strength, $F$ means the line shape function, and $n$ is the number density of the absorber.

Sensor characteristics also have to be taken into account by the forward model, including the antenna field-of-view, the sideband folding, and the spectrometer channel response (Eriksson et al., 2006).

Firstly, the radiance which encounters the antenna response could be expressed by the integration:

$$I_v^a = \int_\Omega I_v(\Omega)W_v^a(\Omega)d\Omega, \tag{4}$$

where $W_v^a$ is the normalized antenna response function. Normally, the variation of $I_v$ in azimuth angle dimension can be neglected or calculated before-hand. Secondly, a heterodyne mixer converts the signals to intermediate frequency, folding the upper and lower sideband signals together in consequence. The apparent intensity after the mixer can be modelled as:

$$I_v^{if} = \frac{W_v^S(v)I_v^a + W_v^S(v')I_{v'}^a}{W_v^S(v) + W_v^S(v')}, \tag{5}$$

where $W_v^S$ means the sideband response. At last, the final signal will be recorded by spectrometers, which can be described

in a similar way as the antenna response:

$$I^c = \int_v I_v^{if} W_v^c(v)dv. \tag{6}$$

Here $W_v^c$ means the normalized channel response, and the radiance is denoted $I^c$.

The measured radiance is transformed to brightness temperatures using the Planck's function.

### 3.2 Retrieval algorithm

Optimal estimation method (OEM) is the most common method used in atmospheric sounding for retrieving vertical profiles of chemistry species (Rodgers, 2000).

In OEM theory, a predicted noisy measurement $\hat{y}$ can be expressed by a forward model $F$ with an unknow atmospheric state $x$ and the system noise $\epsilon_y$ according to:

$$\hat{y} = F(x, b) + \epsilon_y. \tag{7}$$

The noiseless predicted radiance $F(x, b)$ are compared with the observed radiance $y$ so that the unknow state which minimize the cost function $\chi^2$ could be found. The cost function is given by:

$$\chi^2 = [y - F(x, b)]^T S_y^{-1} [y - F(x, b)] + [x - x_a]^T S_a^{-1} [x - x_a], \tag{8}$$

where $x_a$ is a priori state vector, $S_a$ and $S_y$ stand for the covariance matrices representing the natural variability of the state vector and the measurement error vector, respectively. Assuming there is no correlation between channels, the off-diagonal elements of $S_y$ are zero and the diagonal elements are set to the square of the system noise. Usually, a simple formula can be used to determine the SSB radiometric noise standard deviation:

$$\epsilon = \frac{T_{sys}}{\sqrt{\beta \, d\tau}}, \tag{9}$$

where $T_{sys}$ is the system noise temperature which is the sum of receiver noise temperature and the atmospheric temperature received by the antenna, $\beta$ is the noise equivalent bandwidth and $d\tau$ is the integration time for measuring a single spectrum. The diagonal elements of the $S_a$ specify a priori variance and the off-diagonal terms are used to describe correlations between adjacent elements in order to make the retrieved profile smoother. Planck function is used to compute the brightness temperature.

Finally, the Levenberg–Marquardt method which is the modification of the Gauss-Newton iterative is used to solve the nonlinear problem. The solution is given by

$$x_{i+1} = x_i + \left[ (I + \gamma) S_a^{-1} + K_{xi}^T S_y^{-1} K_i \right]^{-1} \left\{ K_{xi}^T S_y^{-1} [y - F(x_i)] - S_a^{-1} (x_i - x_a) \right\}, \tag{10}$$

where $\gamma$ denotes the Levenberg–Marquardt parameter, and $K_{xi}$ represents the weighting function matrix (Jacobian).

The OEM method provides an approach to describe the retrieval error completely. The averaging kernel matrix $A$, which represent the sensitivity of the retrieved state to the true state, is written as:

$$A = G_y K_x = \frac{\partial \hat{x}}{\partial x}, \tag{11}$$

where the $G_y$ is the contribution matrix, which express the sensitivity of the retrieved state to the measurement:

$$G_y = \frac{\partial \hat{x}}{\partial y} = \left( K_x^T S_y K_x + S_a^{-1} \right)^{-1} K_x^T S_y^{-1}. \tag{12}$$

The retrieval resolution can be estimated from the full width at half-maximum (FWHM) of the averaging kernel (Marks and Rodgers, 1993).

There is another useful variable defined as measurement response, which represents the true state contribution in the retrieval (Baron et al., 2002):

$$W(i) = \sum_j |A(i, j)|. \tag{13}$$

The ideal measurement response should be near 1. In practice, reliable range of a retrieval is usually characterised by $|W - 1| < 0.2$.

The retrieval error can be described by two covariance matrices, the smoothing error covariance matrix which is from the need of a priori information:

$$S_n = (A - I)S_a(A - I)^T, \tag{14}$$

the measurement error covariance matrix due to the measurement noise:

$$S_m = G_y S_y G_y{}^T, \tag{15}$$

The error covariance matrix used in following simulation is the total of $S_n$ and $S_m$.

## 4 Measurement performance

### 4.1 Simulation setup

The objective of the simulation is to evaluate the observation performance of TALIS. In this simulation, the forward model Atmospheric Radiative Transfer Simulator (ARTS 2.3) and its corresponding retrieval tool Qpack2 are used (Eriksson et al., 2005; Eriksson et al., 2011). The instrumental setup follows the characteristics of TALIS described in Table 1 and Table 2. The ideal rectangle backend channel response function is used. The simulation antenna patterns of the four radiometers are shown in Fig. 8. The full-width at half-power points of antenna patterns are used in the following simulation.

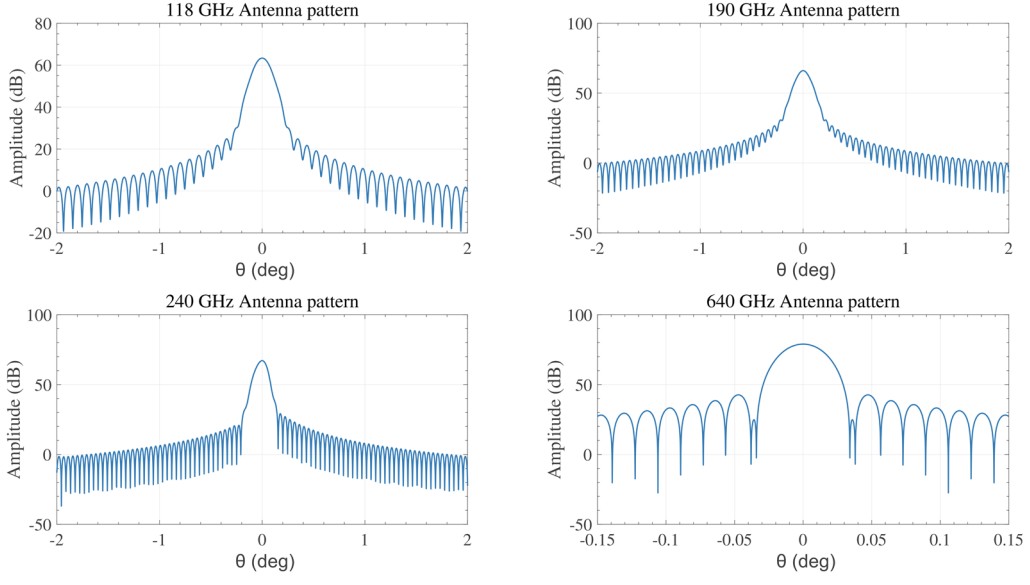

Figure 8. The antenna patterns of TALIS.

In this simulation, the scan altitude range is from 10 to 90 km and the spectra are obtained every 1 km. A retrieval grid with 2.5 km spacing is used since it can match the FOV of TALIS well, and cutting down the size of the state vector will give a significant increase in speed (Livesey and Snyder, 2004). A mid-latitude summer atmospheric condition extracted from FASCOD which is provided by ARTS (profiles of BrO and $HO_2$ are from MLS L3 monthly averaged data, 20°N-30°N, July, 2018) is chosen to perform the simulation. The scattering from tropospheric clouds, refraction, and Zeeman effect are not considered because of the large computational complexity. A spectroscopic line parameters catalogue created with the

data taken from JPL catalogue (Pickett et al., 1998), HITRAN database (Rothman et al., 2013), and Perrin catalogue (Perrin et al., 2005) is used for line-by-line absorption calculation. The measurement covariance matrix is set diagonal as described in Sect.3 in order to reduce the computing time. 110 % of a typical profile is used to build the a priori covariance matrix with 3 km vertical correlation between the adjacent pressure levels by a parametric exponential function. The true profiles are defined with a vertical resolution of 0.5 km. The true species profiles are multiplied by a factor of 1.1 to be the a priori profiles, and the true temperature profile is added a 5 K offset to be the a priori profile. The molecules are retrieved simultaneously from each band.

The expected 1σ noise is calculated by Eq. (9), and the noise is assumed to be 2.2 K, 2.2 K, 2.2 K, and 5.1 K at 118, 190, 240, 640 GHz, respectively. The species such as BrO and $HO_2$ which emission radiances are small compared with the system noise must be averaged to increase the precision. Here the lower noise (1σ noise multiply a factor of 0.1, equivalent to a 10-degree latitude weekly zonal mean) is used to represent the averaged production.

## 4.2 Comparison of TALIS and Aura MLS

As discussed in section 2.2, TALIS has similar bands to Aura MLS. The major difference between these two instruments is the spectrometers used in limb sounding. A simulation is performed to compare the performance of the main products between TALIS FFT spectrometer and Aura MLS 'Standard' 25-channel spectrometer. Figure 9 to 11 show the retrieval products of TALIS and MLS, all the factors are identical except the spectrometer.

According to the simulation results, TALIS can do a better job than Aura MLS because of the wider bandwidth and finer resolution. Temperature precision of TALIS is 1.5 K better than Aura MLS at about 15–30 km and the vertical resolution is also improved. The difference of precision becomes small above 50 km. $H_2O$ precision is improved about 2–10% at about 15–50 km. $O_3$ precision is improved about 3–20% at about 10–60 km. However, the Digital Autocorrelator Spectrometers of MLS which can improve the performance in the mesosphere are not considered here.

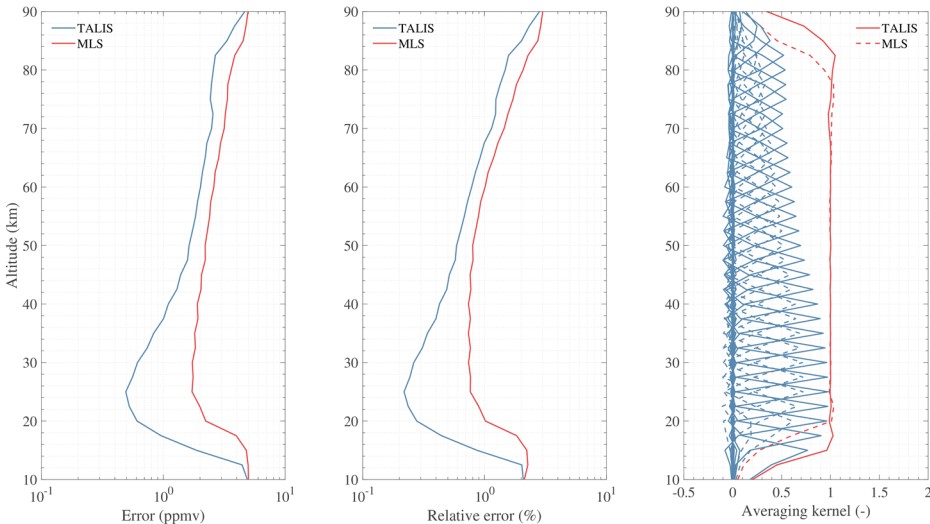

**Figure 9. Temperature product comparison between TALIS FFT spectrometer and MLS 'Standard' spectrometer using 118.75 GHz line. All other factors are identical.**

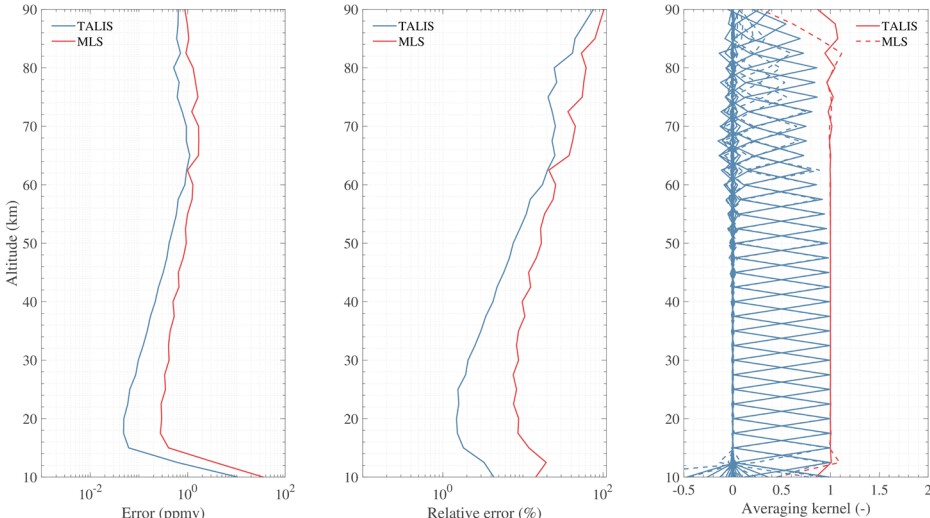

5   **Figure 10. H$_2$O product comparison between TALIS FFT spectrometer and MLS 'Standard' spectrometer using 183.31 GHz line. All other factors are identical.**

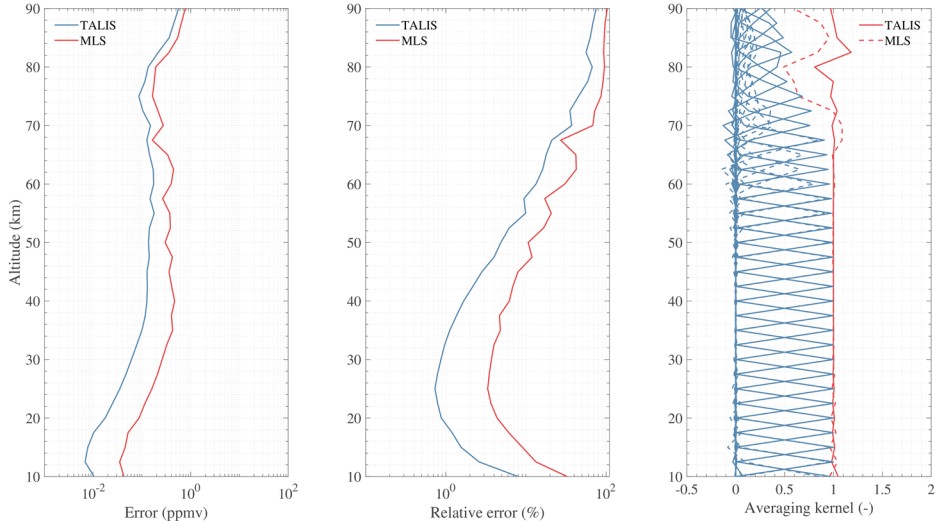

**Figure 11. O₃ product comparison between TALIS FFT spectrometer and MLS 'Standard' spectrometer using 235.71 GHz line. All other factors are identical.**

## 4.3 Retrieval precision

Since the simulation has been performed, an evaluation of the retrieval precision on the target species of TALIS is made. Retrieval profile, a priori profile, and true profile are all plotted in Figs. 12 to 28. The precision (square root of diagonal elements of the error covariance matrix) is given for a single scan and averaged measurement respectively, and the relative error is also provided. Auxiliary information about averaging kernel function and measurement response are also included. Results are discussed in details in the followings.

### 4.3.1 Better precision products

Temperature, $H_2O$, $O_3$, $HNO_3$, HCl, $N_2O$, and ClO are treated as better precision products because of the good precision for a single scan measurement. These products can be used in scientific research directly.

Atmospheric temperature is the most important parameter that can be retrieved with high signal-noise ratio in lower frequency or good vertical resolution in high frequency by using $O_2$ lines. TALIS will use 118 GHz radiometer to detect atmospheric temperature profile, with 240 and 643 GHz radiometers worked as supplement products. Results are shown in Fig. 12, the sensitivity is significantly high at the 118 GHz band. Single scan precision is good from 15 to 60 km with the precision < 2 K. The retrieval vertical resolution is 2.5-4 km below 50 km and 4–6 km from 50 to 80 km. The precision of averaged measurement will be < 1 K from 15 to 85 km.

'Wide' filters of MLS make measurements extending down into the troposphere where TALIS lacks sensitivity. However, the 240 GHz product can compensate for the loss of information since the precision is better in the upper troposphere (error

< 1 K for a vertical resolution of 2.5-3 km between 10 and 15 km). Result of 643 GHz band is similar to that of 240 GHz band.

Once the temperature profile is retrieved, the pressure profile can be calculated from the hydrostatic equilibrium equation using a known pressure and temperature at a reference tangent point. The pressure profile is not a direct product and is not shown here.

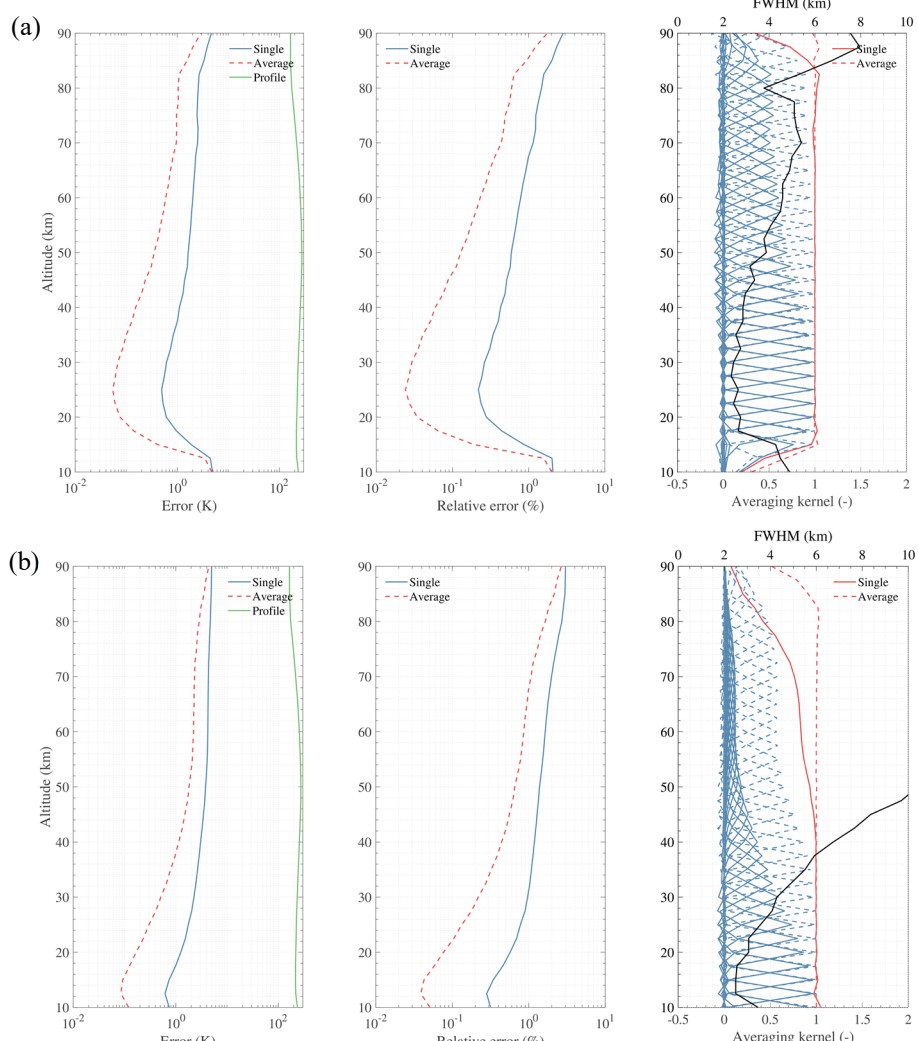

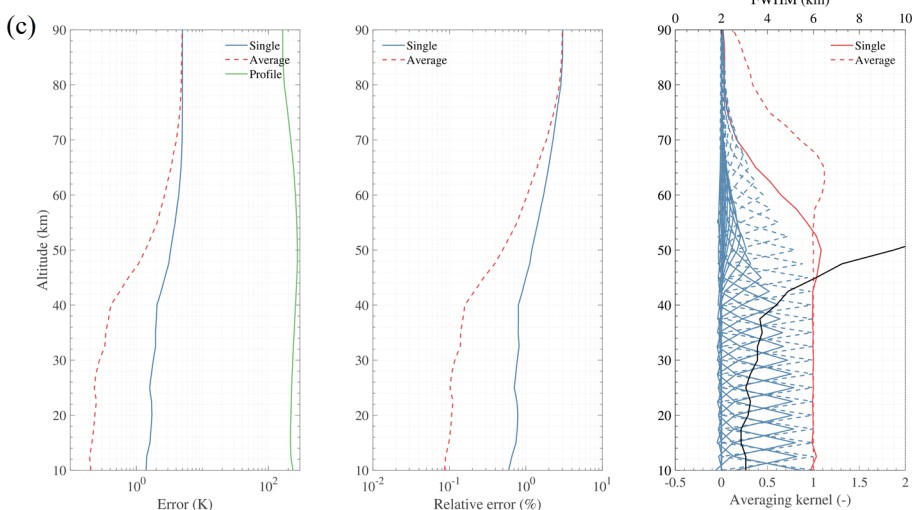

**Figure 12. Simulation results of temperature retrieval using 118.75 (a), 233.95 (b), and 627.75 GHz (c) lines. Single and Average represent the retrieval error using different noise. Profile represent the typical profile used in simulation. The black solid line in the last panel represent the FWHM (i.e. vertical resolution).**

5     The $H_2O$ profile, another key parameter, can be measured by 190 and 643 GHz radiometers. The 183.31 GHz line is generally used by humidity sounder to detect water vapour with good precision. Figure 13 shows the retrieval precision will be < 10% from 10 to 55 km by 190 GHz single scan measurement with the vertical resolution of 2.5–4 km. Averaged measurement has the retrieval precision < 1% at 10–55 km, < 5% at 10–80 km. The profile can also be retrieved by 643 GHz radiometer with poorer precision.

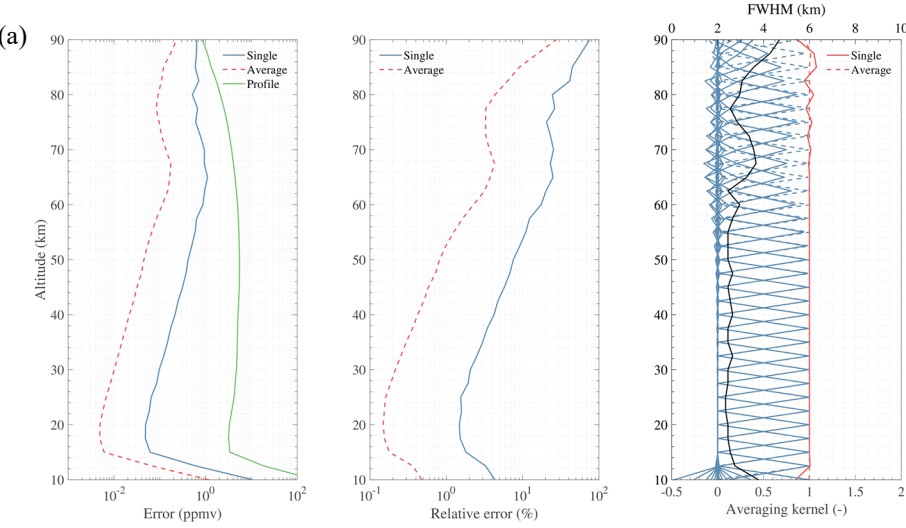

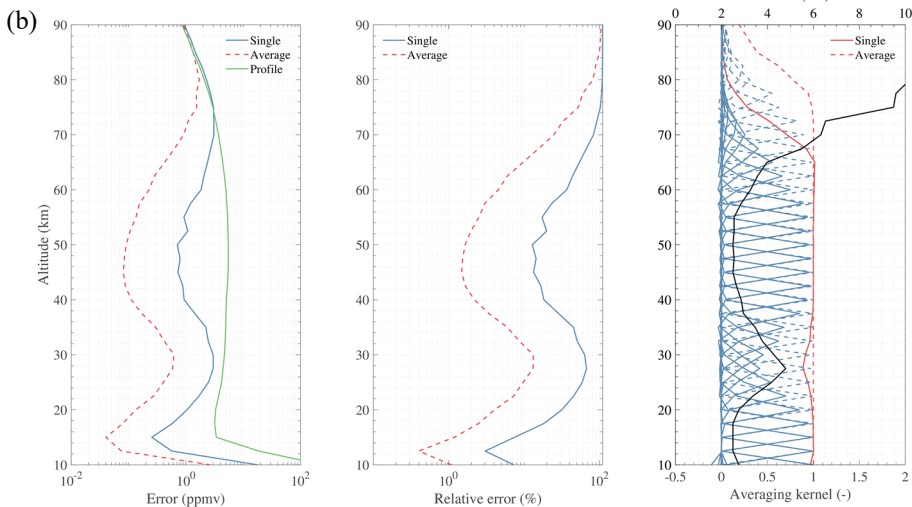

**Figure 13. Simulation results of H₂O retrieval using 183.31 (a) and 657.9 GHz (b) lines. Single and Average represent the retrieval error using different noise. Profile represent the typical profile used in simulation. The black solid line in the last panel represent the FWHM (i.e. vertical resolution).**

5    O₃ has quite strong intensity in most spectral regions of TALIS. All the radiometers except 118 GHz can be used to observe this gas which is important for energy balance (Fig. 14). The 240 GHz radiometer which covers the 235.7 GHz line has the highest O₃ sensitivity. The profile can be retrieved with a single scan precision < 10% from 10 to 55 km and the vertical resolution is 2.5–3 km. The vertical resolution will degrade to 3–6 km for altitudes higher than 70 km. By averaging the measurements, the precision will be < 5% at 10–70 km. The other two bands show good performance from 15 to 55 km

10   with a single scan precision < 10%.

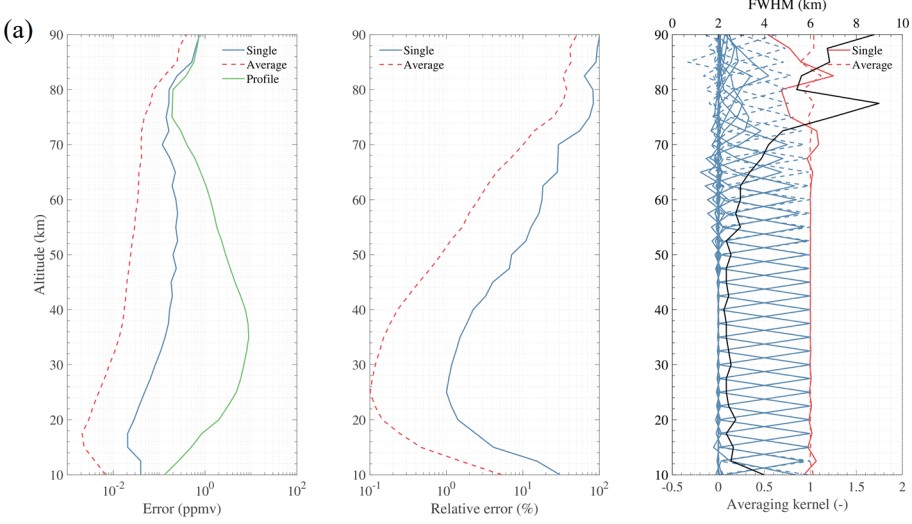

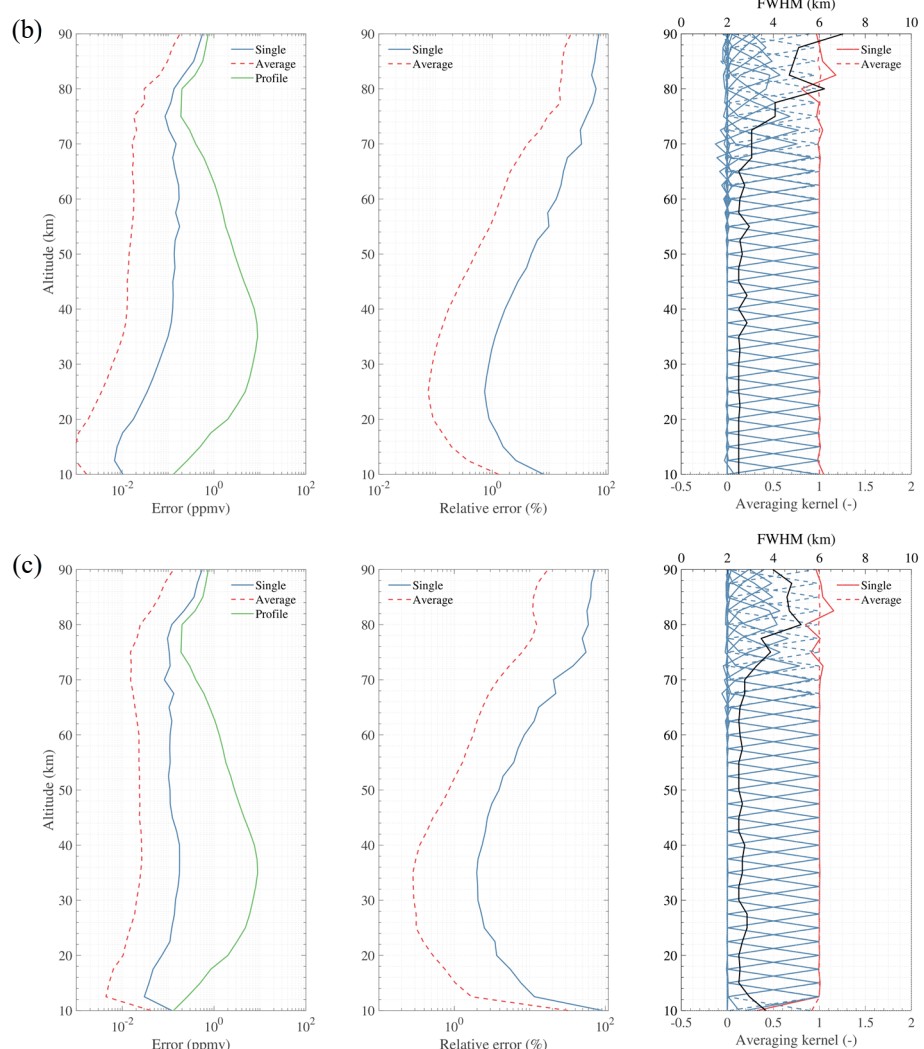

**Figure 14. Simulation results of O₃ retrieval using 190 (a), 235.7 (b), and 657.5 GHz (c) lines. Single and Average represent the retrieval error using different noise. Profile represent the typical profile used in simulation. The black solid line in the last panel represent the FWHM (i.e. vertical resolution).**

HNO₃ is a common species in the stratosphere and has relatively strong lines at 240 and 643 GHz bands. Figure 15 shows the results of HNO₃ retrieval. The 240 GHz radiometer can measure HNO₃ at 15–32 km altitude range with a single scan precision < 30% and the vertical resolution is 2.5–3 km. Averaging the measurements can improve the retrieval with a precision < 10% from 15 to 35 km. The 643 GHz signal is stronger than that in the 240 GHz band, but it is strongly absorbed by O₃ below about 30 km. However, after averaging the measurements, information can be retrieved between 15 and 70 km with a precision better than 60%.

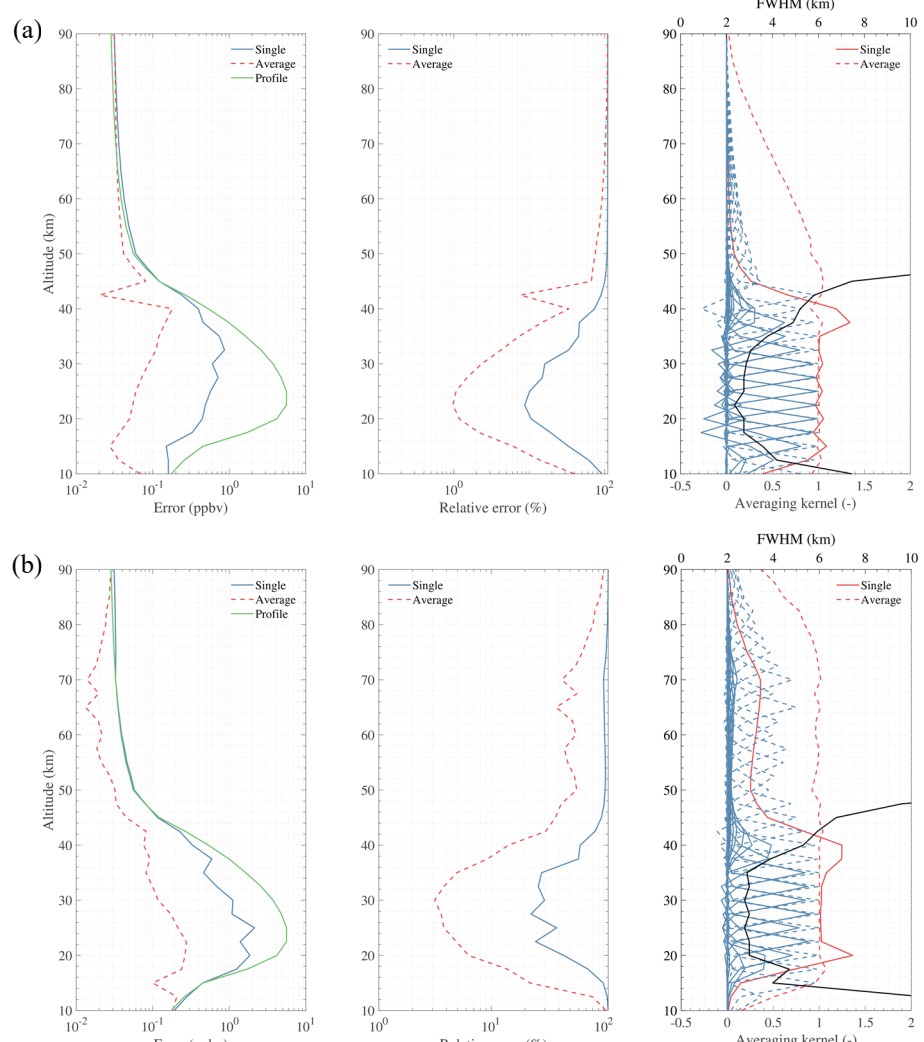

**Figure 15. Simulation results of HNO₃ retrieval using 244 (a), and 656 GHz (b) lines. Single and Average represent the retrieval error using different noise. Profile represent the typical profile used in simulation. The black solid line in the last panel represent**
5    **the FWHM (i.e. vertical resolution).**

Figure 16 shows the expected precision of HCl observation. HCl can be measured at 15–50 km with < 20% single scan relative error with the vertical resolution of 2.5–3 km. By averaging the measurements, the precision will be < 10% at 12–72 km.

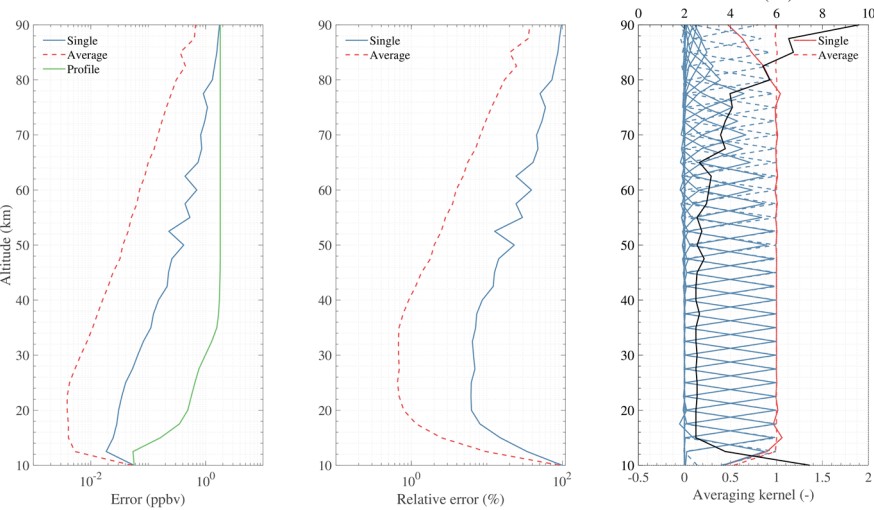

**Figure 16. Simulation result of HCl retrieval using 624.9 GHz lines. Single and Average represent the retrieval error using different noise. Profile represent the typical profile used in simulation. The black solid line in the last panel represent the FWHM (i.e. vertical resolution).**

N₂O can be retrieved from the band at 190 GHz in the upper troposphere while the band at 643 GHz can provide more information and good precision in the stratosphere. Figure 17 shows that single scan precision of 190 GHz is < 20% at 12–32 km with the vertical resolution of 2.5 km. By averaging the measurements, the precision will be < 10% from 10 to 42 km. The 190 GHz can give the similar precision at 10–20 km.

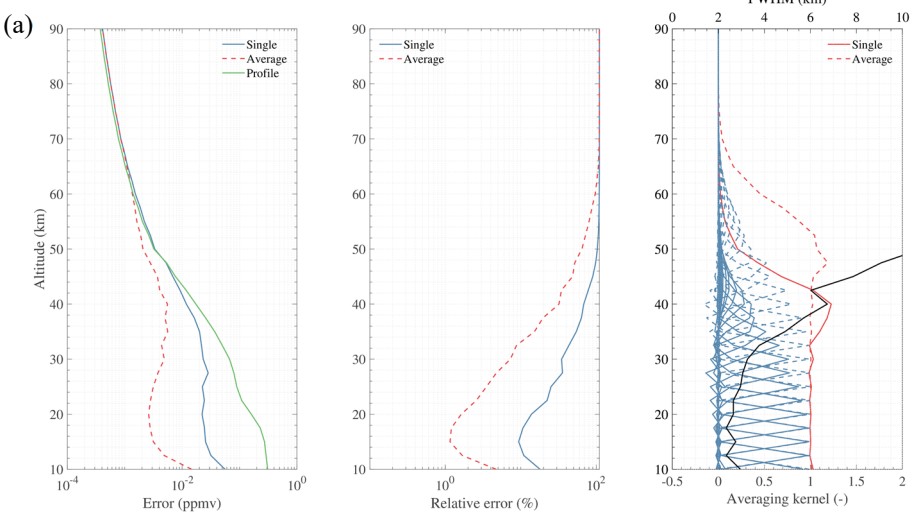

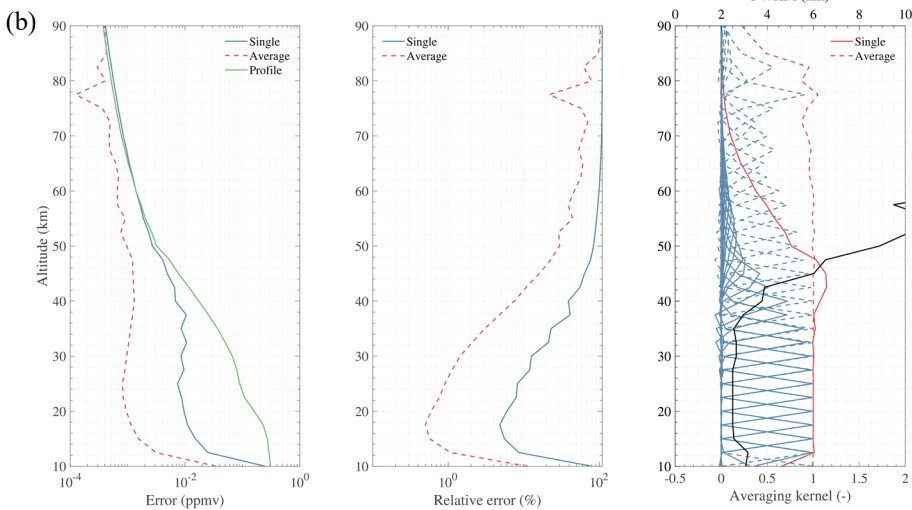

**Figure 17. Simulation results of N₂O retrieval using 200.98 (a), and 652.834 GHz (b) lines. Single and Average represent the retrieval error using different noise. Profile represent the typical profile used in simulation. The black solid line in the last panel represent the FWHM (i.e. vertical resolution).**

5    ClO can be retrieved from radiances measured by 190 and 643 GHz bands (Fig. 18). However, the result shows that the best retrievals are performed from the band at 643 GHz but information can also be retrieved from the 190 GHz radiometer with poorer precision. Single scan measurement from 643 GHz radiometer can be used to obtain ClO with < 40% precision from 30 to 45 km, and the vertical resolution is about 2.5–4 km throughout the useful range. By averaging the measurements, precision will be < 30% from 23 to 57 km. Since ClO will vanishes in the middle stratosphere (30-40 km) during nighttime,

10   the precision will be worse in the nighttime. In the polar regions, the relative precision will be better between 20 and 25 km during chlorine activation.

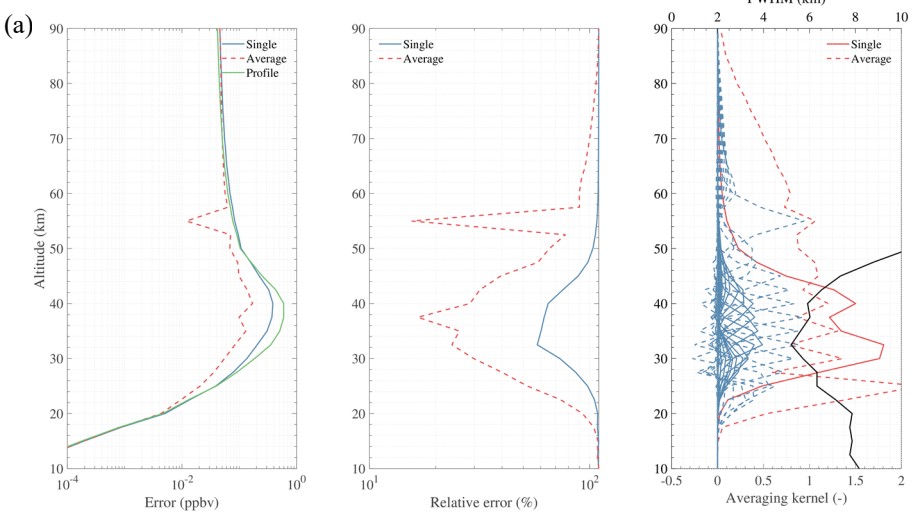

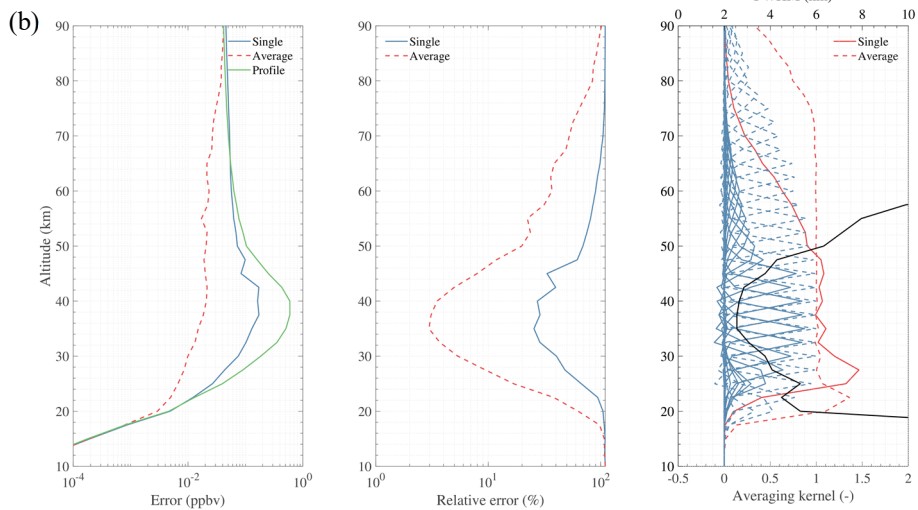

**Figure 18. Simulation results of ClO retrieval using 203.4 (a) and 649.45 GHz (b) lines. Single and Average represent the retrieval error using different noise. Profile represent the typical profile used in simulation. The black solid line in the last panel represent the FWHM (i.e. vertical resolution).**

## 4.3.2 Medium precision products

Medium precision products including CO, HCN, CH₃Cl mean that their single scan retrieval precisions are not satisfying, but can be used to some degree. There is a choice for the user to select the single scan or averaged products.

CO can be measured using 230.538 and 661.07 GHz lines. Figure 19 shows that the 240 GHz radiometer can provide CO information with 30–90% single scan precision from 10 to 90 km. The vertical resolution is in the range 3.5–5.5 km from the upper troposphere to the lower mesosphere, degrading to 6–10 km in the upper mesosphere. By using averaged measurements, CO can be retrieved with < 30% relative error at the range of 10–90 km. However, the retrieval of 643 GHz measurement shows poor precision.

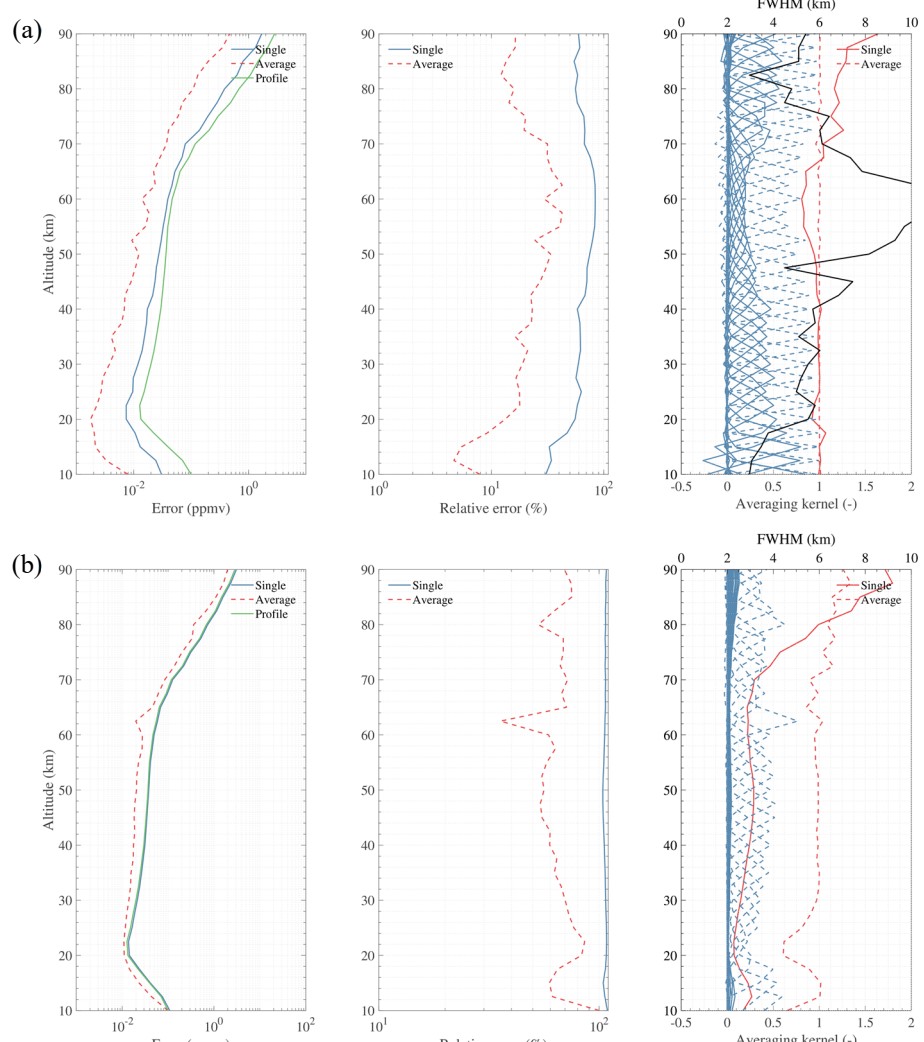

**Figure 19. Simulation results of CO retrieval using 230.538 (a) and 661.07 GHz (b) lines. Single and Average represent the retrieval error using different noise. Profile represent the typical profile used in simulation. The black solid line in the last panel represent the FWHM (i.e. vertical resolution).**

HCN is measured by 190 GHz radiometer at 177.26 GHz line. The single scan precision is < 50% from 12 to 28 km and the vertical resolution is about 5 km at the height of 30 km, degrading to 8 km at about 40 km (Fig. 20). By averaging the measurements, the relative error will be < 30% at 10–40 km.

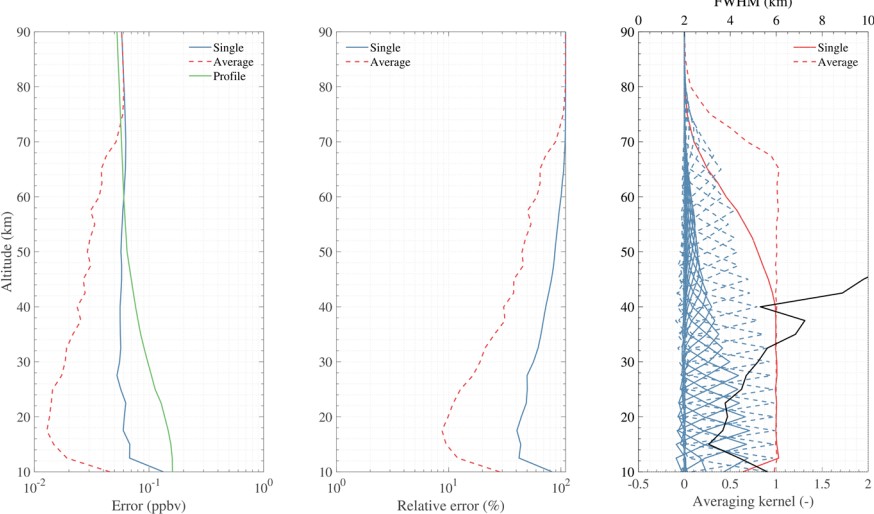

**Figure 20. Simulation result of HCN retrieval using 177.26 GHz line. Single and Average represent the retrieval error using different noise. Profile represent the typical profile used in simulation. The black solid line in the last panel represent the FWHM (i.e. vertical resolution).**

5    CH₃Cl can be measured by the 643 GHz radiometer. As the result shows (Fig. 21), the 649.5 GHz band are suitable for CH₃Cl observation in the upper troposphere and lower stratosphere. It can be measured with < 30% single scan precision from 12 to 23 km, with < 20% averaged precision from 10 to 30 km. The vertical resolution is about 3 –4 km over most of the useful range.

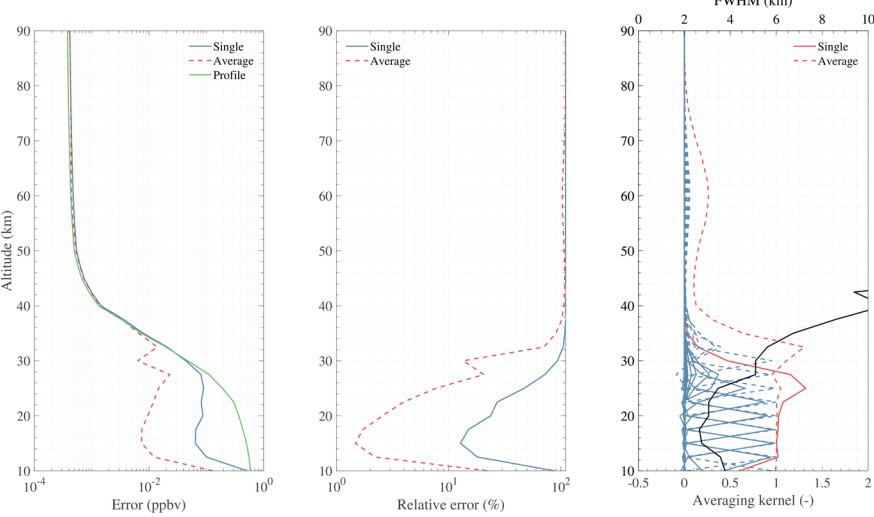

10    **Figure 21. Simulation results of CH₃Cl retrieval using 649.5 GHz lines. Single and Average represent the retrieval error using different noise. Profile represent the typical profile used in simulation. The black solid line in the last panel represent the FWHM (i.e. vertical resolution).**

### 4.3.3 Poor precision products

There are several weak lines in the spectral regions of TALIS such as HOCl, BrO, and HO$_2$. Significant averaging must be done to these measurements in order to obtain reliable and satisfying precision.

The 635.87 GHz line is the most appropriate line for HOCl observation. However, the single scan retrieval has poor precision of 60–80% at 25–45 km with the vertical resolution of about 4–6 km. Figure 22 reveals that HOCl can be retrieved from 20 to 50 km with averaged measurement precision of < 50%.

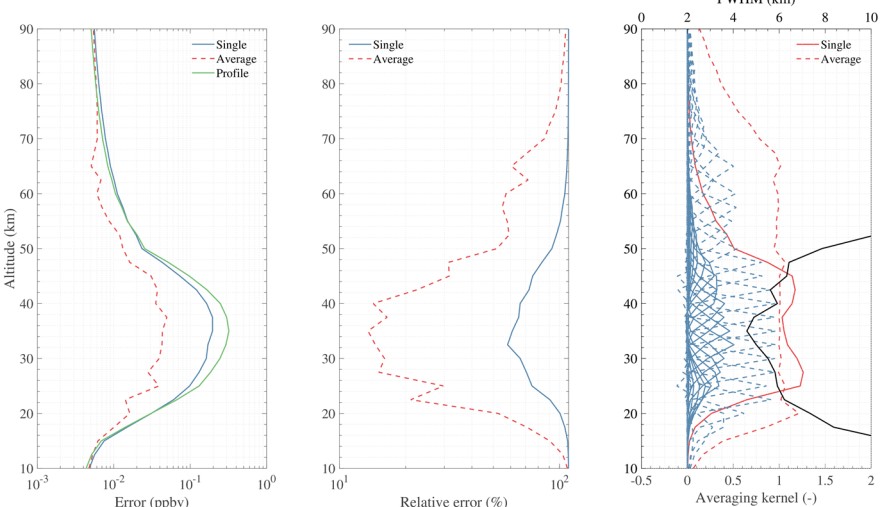

**Figure 22. Simulation result of HOCl retrieval using 635.87 GHz line. Single and Average represent the retrieval error using different noise. Profile represent the typical profile used in simulation. The black solid line in the last panel represent the FWHM (i.e. vertical resolution).**

BrO can be measured by using 624.768 GHz spectral line. Figure 23 shows the simulation result of BrO retrieval. As the averaging kernel reveals, there is almost no useful information in single scan measurement because of the quite poor signal-to-noise ratio. Therefore, averaging is needed to obtain reliable and scientific results. The error is 50% from 24 to 48 km with the vertical resolution of about 4 km.

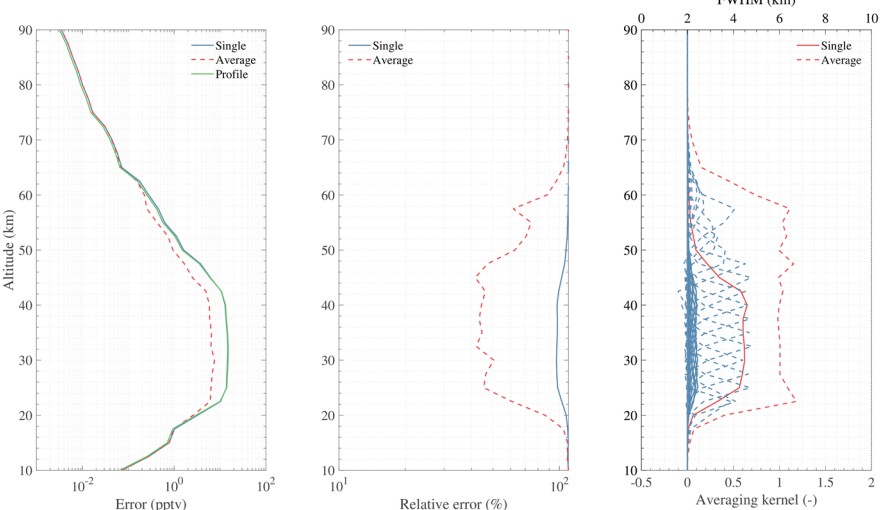

**Figure 23. Simulation results of BrO retrieval using 624.768 GHz line. Single and Average represent the retrieval error using different noise. Profile represent the typical profile used in simulation. The black solid line in the last panel represent the FWHM (i.e. vertical resolution).**

HO₂ can be measured by the 643 GHz radiometer with < 50% precision at the vertical range of 30–90 km by using averaged data (Fig. 24). The precision of single scan retrieval is 55–70% at 40–75 km which is not desirable because of the weak signal. The vertical resolution is about 6 km.

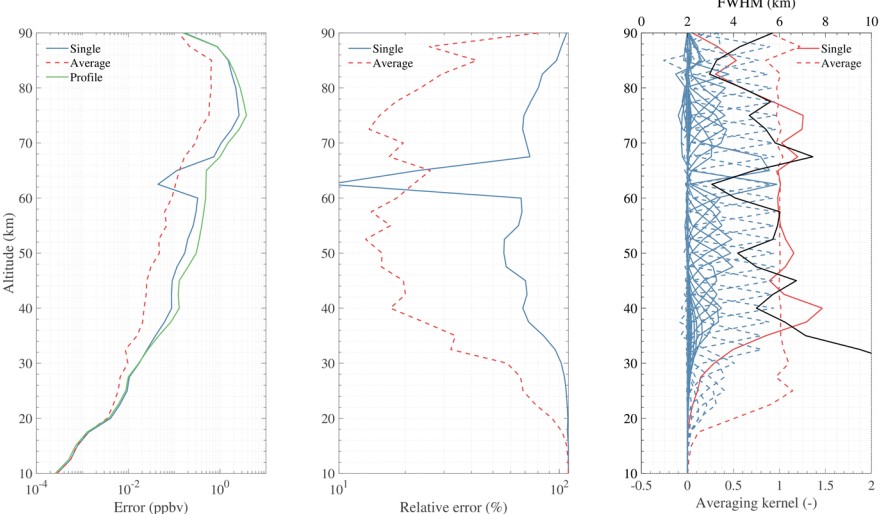

**Figure 24. Simulation results of HO₂ retrieval using 649.701 GHz line. Single and Average represent the retrieval error using different noise. Profile represent the typical profile used in simulation. The black solid line in the last panel represent the FWHM (i.e. vertical resolution).**

### 4.3.4 Promising products

The unique products are the target species which are not covered by Aura MLS but covered by TALIS. There are four gases: NO, $NO_2$, $H_2CO$, and $SO_2$ (normal VMR). However, their signals all have weak intensity and must be averaged to improve the retrieval precision.

NO (daytime) can be retrieved from averaged data with < 50% precision at 28–90 km (Fig. 25) while it vanishes in the nighttime. The vertical resolution is about 4–10 km. While its single scan measurement has no information in the area where NO largely exists.

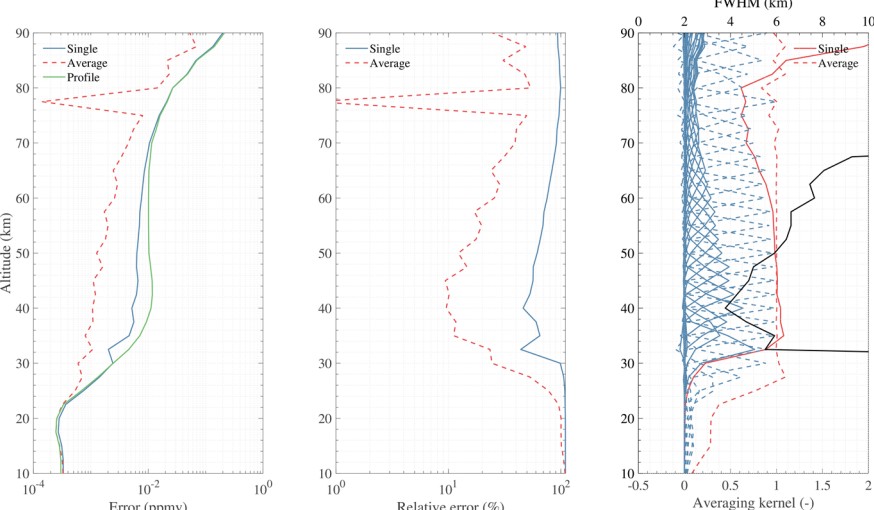

**Figure 25. Simulation result of NO retrieval using 651.75 GHz line. Single and Average represent the retrieval error using different noise. Profile represent the typical profile used in simulation. The black solid line in the last panel represent the FWHM (i.e. vertical resolution).**

$NO_2$ (nighttime) has a weak line in the spectrum of 240 GHz band and it vanishes in the daytime. Figure 26 shows that only averaged measurement can provide some information at 20–40 km with the precision of about 40–60 % in the nighttime. The vertical resolution is about 5 km.

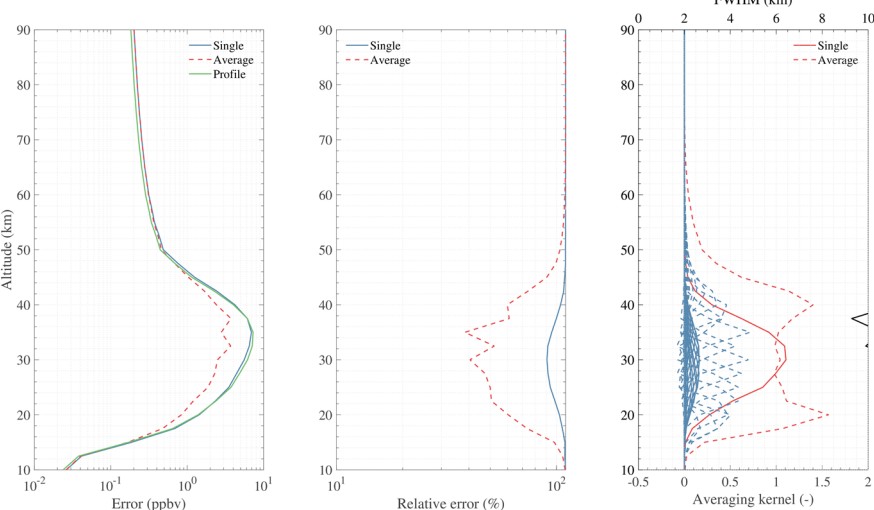

**Figure 26. Simulation result of NO₂ retrieval using 232.7 GHz lines. Single and Average represent the retrieval error using different noise. Profile represent the typical profile used in simulation. The black solid line in the last panel represent the FWHM (i.e. vertical resolution).**

5    Although H₂CO has a line at 656.45 GHz, its emission radiance is too weak. Almost no useful information can be obtained (Fig. 27). However, this line has the potential to measure H₂CO. More average or other effective methods should be applied to get acceptable precision.

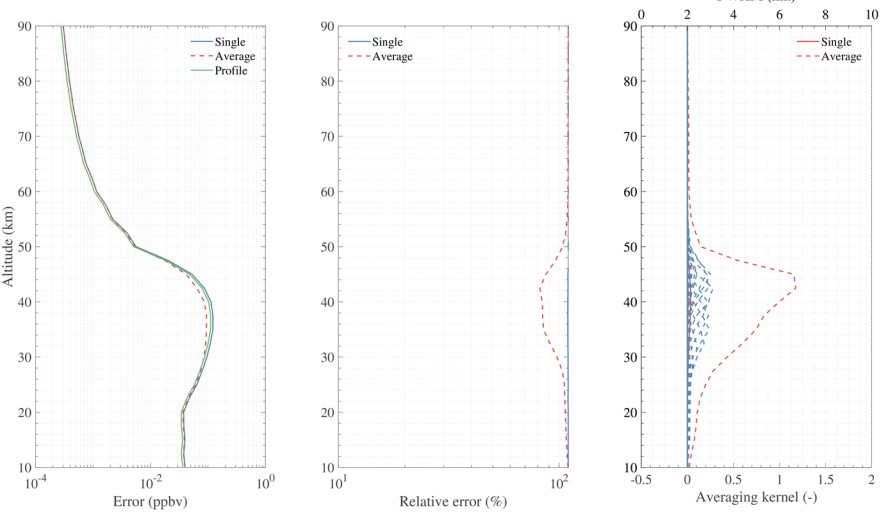

**Figure 27. Simulation result of H₂CO retrieval using 656.45 GHz line. Single and Average represent the retrieval error using**
10    **different noise. Profile represent the typical profile used in simulation. The black solid line in the last panel represent the FWHM (i.e. vertical resolution).**

MLS standard $SO_2$ product is taken from the 240 GHz retrieval, but only effective when its concentration significantly enhanced. TALIS has both 240 and 643 GHz radiometer which covering the lines of $SO_2$. The 240 GHz radiometer can be used to measure $SO_2$ like the way of MLS. The 643 GHz radiometer can give the concentration of nominal background. The averaged result shows that $SO_2$ can be retrieved at 14–20 km, 46–70 km with the relative error about < 50% (Fig. 28). The vertical resolution is about 6 km.

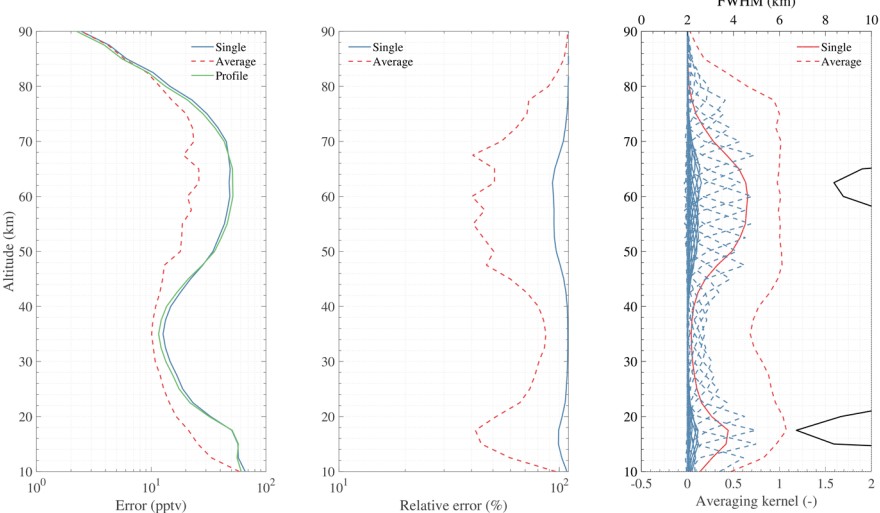

**Figure 28. Simulation result of $SO_2$ retrieval using 659 GHz lines. Single and Average represent the retrieval error using different noise. Profile represent the typical profile used in simulation. The black solid line in the last panel represent the FWHM (i.e. vertical resolution).**

## 5 Conclusions

Simulation analysis for temperature and chemical species retrieval have been performed to assess the measurement performance of TALIS and to support the mission. This study mainly focuses on a large number of important chemical species in middle and upper atmosphere which can be observed by limb sounder. The results are summarized in Table 3.

Seven species show high sensitivity, sufficient for scientifically useful single profile retrievals. 118, 240, and 643 GHz observations of $O_2$ are used to estimate temperature profile which is quite important in meteorology. The 118 GHz radiometer can obtain temperature with a precision < 2 K at 15–80 km and the 240 and 643 GHz radiometers can provide information in the upper troposphere. The 190 GHz radiometer can be used to measure $H_2O$ with a precision < 10% in a wide vertical range and give information of upper tropospheric humidity. $O_3$ can be measured by three radiometers, and the 240 GHz radiometer has the best precision. The precision is < 5% from 25 to 60 km by single scan measurement. $HNO_3$ can be derived from 240 GHz retrieval with a precision < 40%. The precision of HCl single scan retrieval is < 40% over most of the useful range. The 190 GHz radiometer can give a good estimate to $N_2O$ profile with a precision about 20–40%, while 643

GHz measurement can provide more information at higher altitudes. Single scan precision of ClO measured by 643 GHz radiometer is about 30% in the area where ClO mainly exists. $CH_3Cl$ can be measured in the upper troposphere and low stratosphere with a precision about 50%. The profile of CO retrieved from 240 GHz measurement is better than that from 643 GHz measurement. The best sensitivity is found between 70~90 km where the VMR of CO is large, and the precision is better than 50%. HCN have 50% single scan precision at 18–32 km which may need to be averaged. Other measurements such as $HO_2$, HOCl, NO, $NO_2$, BrO, $SO_2$, and $H_2CO$ must be significantly averaged before scientific use because of the weak signals. As the results show that the precision of some products become much poorer suddenly at a height of 25 km, it seems that the retrieval grid resolution does not match the achievable resolution.

Apart from these products, some potential products will be discussed in the future works. Line-of-sight wind is important information which could be measured by TALIS. Cloud IceWater Content (IWC) is also an essential product provided by passive microwave radiometer. Future studies will also investigate the Zeeman effect as it polarizes and changes the shape of the $O_2$ lines.

TALIS has strong potential to monitor chemical composition in the whole Earth's atmosphere which is important for numerical weather prediction models and to characterize the long-time change of climate. Measurement data can be used for atmospheric chemistry and dynamics study which is quite important for the geoscience. A better understanding of the key chemical and dynamical processes in the middle and upper atmosphere will help us solve the climate problem more efficient.

This paper is the preliminary analysis of the instrument. More studies such as structure optimization, calibration research, and error analysis will be performed to support the mission.

**Table 3. Simulation results of TALIS retrieval precision**

| Product | Radiometer | Single precision | Average precision |
|---|---|---|---|
| Temperature | 118 GHz, 240 GHz | < 2K (10–60 km) | < 2K (10–85 km) |
| $H_2O$ | 190 GHz | < 10% (10–55 km) | < 5% (10–80 km) |
| $O_3$ | 240 GHz | < 10% (10–55 km) | < 5% (10–70 km) |
| HCl | 643 GHz | < 20% (15–50 km) | < 10% (12–72 km) |
| $N_2O$ | 643 GHz | < 20% (12–32 km) | < 10% (10–42 km) |
| $HNO_3$ | 240 GHz | < 30% (15–32 km) | < 10% (15–35 km) |
| ClO | 643 GHz | < 40% (30–45 km) | < 30% (23–57 km) |
| CO | 240 GHz | 30–90% (10–90 km) | < 30% (10–90 km) |
| HCN | 190 GHz | < 50% (12–28 km) | < 30% (10–40 km) |
| $CH_3Cl$ | 643 GHz | < 30% (12–23 km) | < 20% (10–30 km) |
| HOCl | 643 GHz | 60–80% (25–45 km) | < 50% (20–50 km) |
| BrO | 643 GHz | / | < 50% (24–48 km) |

| | | | |
|---|---|---|---|
| HO$_2$ | 643 GHz | < 70% (40–75 km) | < 50% (30–90 km) |
| NO | 643 GHz | 50–70 % (33–55 km) | < 50% (28–90 km) |
| NO$_2$ | 240 GHz | / | 40–60% (20–40 km) |
| H$_2$CO | 643 GHz | / | / |
| SO$_2$ | 643 GHz | / | < 50% (14–20, 46–70 km) |

*Code and data availability*. ARTS can be downloaded at http://www.radiativetransfer.org/getarts/. Qpack is included in the Atmlab which can be downloaded from http://www.radiativetransfer.org/tools/. Profiles and spectroscopy data of Perrin and HITRAN are included in ARTS XML Data. JPL molecular spectroscopy catalogue is available at https://spec.jpl.nasa.gov/.
MLS version 4.2 data can be obtained at https://doi.org/10.5067/Aura/MLS/DATA3020.

*Author contribution*. Zhenzhan Wang designed the mission concept. Wenyu Wang performed the simulate and wrote the manuscript. Wenyu Wang and Yongqiang Duan analysed the results. Zhenzhan Wang edited the article.

*Competing interests*. The authors declare that they have no conflict of interest.

*Acknowledgements*. The authors would like to thank the ARTS and Qpack development teams for assistance configuring and running the model. The authors thank the JPL for providing spectroscopy data and MLS data. They would also like to thank the reviewers and the editors for their valuable and helpful suggestions.

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
