# Peer review of "Performance Evaluation of THz Atmospheric Limb Sounder (TALIS) of China"

_Atmospheric Measurement Techniques, 2019_

## Referee Comment (RC1) · Hugh C. Pumphrey (Referee) · 15 Jul 2019

**1   General Comments**

This paper describes the performance of a proposed instrument. If flown, the instrument will make an important contribution to monitoring the chemistry of the middle atmosphere, particularly in view of the fact that EOS MLS on NASA's Aura satellite will inevitably cease operation in the next few years, and no similar instrument other than TALIS is planned to continue the EOS MLS record. The paper should be made available to the public in some way for that reason. I am not entirely convinced that it deserves to be published in AMT, though, and certainly not as it stands. The reason for this is that the proposed instrument is quite close to being a carbon copy of the EOS

[Figure]

MLS instrument. EOS MLS has been in flight for 15 years, now, and its performance and limitations have been studied and reported in enormous detail. A simulation of the performance of an instrument which is very similar to EOS MLS provides only a very small advance in knowledge.

Where the two instruments differ is in the spectrometers used. EOS MLS has old-fashioned filter banks, whereas TALIS will use FFT spectrometers with many more channels. For this paper to demonstrate any novelty, it needs to show how the coverage of the TALIS spectrometers differs from that of the MLS filter banks, and to demonstrate the extent to which the improved coverage leads to improved quality of retrieval products. This would involve simulating retrievals with the two sorts of spectrometers, with all other factors kept identical.

The simulation reported has a number of failings. The most serious of these are the failure to properly consider the vertical resolution of the instrument, and the failure to describe the antenna characteristics assumed; these characteristics are the main factor limiting the vertical resolution of the instrument. For the paper to be published, the antenna characteristics should be described, and the inevitable tradeoff between resolution and precision considered properly. Another problem is the failure to address how information on the geometrical tangent height is to be incorporated into the retrieval of temperature and tangent pressure.

The paper is generally presented adequately for the most part. The standard of written English is rather variable, but is such that the authors' meaning is always clear. The figures are of a good standard in most respects, but I make some suggestions for improvement below.

**2 Specific comments**

- Page 1 line 18: In "high single scan retrieval precision of 1 – 50%", remove the word "high". 1% can be considered high precision in this context, but 50% is not high precision in any context. I suppose you could replace "high" with "relatively high".

- Page 1 line 25–27: It seems odd to talk of this instrument as a "Terahertz limb sounder" when its highest frequency is 0.6 THz and its lowest frequency is below 0.2 THz.

- Page 3 lines 5–7: The double sideband nature of both UARS MLS and Aura MLS has been a considerable nuisance to the instrument team, especially when attempting to obtain results in the important 300 hPa – 60 hPa range. It was a technological limitation at the time EOS MLS was built, which would have been too expensive to work around. Modern mm-wave technology includes sideband-separating receivers. As TALIS appears still to be in the design phase, the authors might like to consider whether this technology would be appropriate.

- Page 3 line 14: "TALIS covers most spectral bands of EOS MLS and extends them." It would perhaps be worth adding some sort of diagram with the EOS MLS and TALIS spectral bands overlaid on each other, so that the reader can quickly see by how much the TALIS coverage extends that of EOS MLS. The should also be a statement that EOS MLS had a band at 2.4 THz which TALIS will lack — there is no need of a diagram to show this.

- Page 3 line 15: "... and lower noise of TALIS". The authors should probably state the $T_{\mathrm{sys}}$ values of the EOS MLS bands so that the reader can easily see how much lower the noise level of TALIS will be. A cross-check against Waters et al. (2006) (see Table 1) indicates that the TALIS $T_{\mathrm{sys}}$ values are either effectively the same as those of EOS MLS or are about 30%–50% better. This is unlikely

**Table 1.** System temperature values for TALIS and EOS MLS

| Band | $T_{\mathrm{sys}}$ (TALIS) | $T_{\mathrm{sys}}$ (MLS) |
|---|---|---|
| 118 GHz | 800 | 1200 |
| 190 GHz | 1000 | 900 – 1100 |
| 240 GHz | 1000 | 1300 – 1300 |
| 643 GHz | 3000 | 4000 – 4400 |

to be sufficient to allow easy measurement of a species which EOS MLS can not measure. If TALIS can really do a better job than EOS MLS, it is likely to be on account of the bandwidth and resolution of the spectrometers rather than because of the slightly better measurement noise.

- Page 4 Line 10: It is worth pointing out that the only $SO_2$ which is observable by instruments like EOS MLS and TALIS comes from volcanic eruptions.

- Page 4 Lines 12–13: "...MLS demonstrated that $SO_2$ can be measured by 240 GHz radiometer cooperated with 190 GHz radiometer ...". What I actually said in Pumphrey et al. (2015) is that MLS measures $SO_2$ from three radiometers: 190 GHz, 240 GHz and 640 GHz. The radiometers are not combined with each other. Rather, a separate $SO_2$ product is produced from each radiometer. Only the 240 GHz $SO_2$ product is recommended for general use.

- Page 4 line 14: "$NO_2$ is a unique species not covered by EOS MLS". This is entirely true. The authors should perhaps explain whether TALIS's ability to measure this species is due to improved bandwith, resolution or radiometer noise. Figure 3 suggests that the measurement will be very difficult.

- Page 8 line 5: This formula defines the Planck brightness temperature. It is not uncommon in microwave remote sensing (especially in limb sounding) to work with the Rayleigh-Jeans brightness temperature because it is proportional to the

radiance. The authors should be clear which brightness temperature they intend to use.

- Page 10 section 4.1: This section omits a number of important items. Firstly, it does not state what antenna pattern is assumed when applying equation (5). This is important because the antenna pattern is the main limitation on vertical resolution. The pattern does not always vary with frequency in the way one would naively expect. The 640 GHz band of EOS MLS has better vertical resolution than the 190 GHz band, but nowhere near the three times better that you would expect from Rayleigh's criterion. This is because the 640 GHz radiometer does not view the full aperture of the antenna. Secondly, nothing is said about the specific problem of temperature/pressure retrieval. We usually have two sources of information about where the antenna is pointing: pressure information that comes from the radiance measurements, and geometric height information which comes from the satellite navigation system and the antenna view angle. These two pieces of information are linked by the hydrostatic equation, which itself depends on the temperature profile. For the temperature retrieval precisions presented to be credible, the reader needs to know how the geometric tangent height information was incorporated, and how precise it was assumed to be.

- Page 14ff, figures 6–10: The precision becomes very much poorer very suddenly at a height of 25 km. The authors should explain why this is. The figures make it appear as if the retrieval grid changes vertical resolution at this point from a grid which is coarser than the achievable vertical resolution to one which is finer than the achievable vertical resolution. This will inevitably make the retrieval precision appear far worse below 25 km, but this does not mean that the performance of the instrument itself is far worse below 25 km. The authors should perhaps try showing averaging kernels calculated for a far finer grid, so that the true vertical resolution of the instrument can be assessed. They could then choose a retrieval grid which matches the achievable resolution better.

- Page 23 figure 12: The mixing ratio of CO changes over a very large range. In my experience, the only way to show a vertical profile over the 10 km–90 km range is to use a logarithmic mixing ratio scale.

**3 Technical corrections**

- Page 2 Line 2: "Earth' " should be "Earth's".

- Page 2 line 28: "Spectrum resolution" should be "spectral resolution".

- Page 3 line 12: "criterions" should be "criteria". Although there is a trend in modern English away from using Greek or Latin-derived plurals, "criteria" is in very general use, but nobody says "criterions".

- Page 5, figures 1 – 4: The different lines on these figures can be quite hard to distinguish and to match up with the legend. It would help if the authors were to make the lines slightly thicker and to ensure that they choose strongly-contrasting colours. (They should continue to avoid pure yellow (#ffff00) and pure green (#00ff00) as these colours can be hard to see on a white ground.) The vertical scale currently goes from 1 K to 1000 K, but the data do not cover this entire range. If the software used will permit, the vertical scale should be reduced to cover 2 K to 300 K

- Page 4 line 16: "as more spectral" should be "as many spectral"

- Page 4 lines 24–25 and throughout the paper: in LaTeX, use a non-breaking thin-space (`\,`) between a number and its unit in order to avoid a line break at that point. Here, write `635.87\,GHz`.

- Page 8 line 3: T should be in math mode so that it comes out in math italic ($T$). In LaTeX, write `$T$`, not `T`.

- Diaeresis out of position over de la Nöe's name. Also, "la" does not have a capital letter.

---

## Referee Comment (RC2) · Anonymous Referee #2 · 7 Aug 2019

In the paper "Performance Evaluation of THz Atmospheric Limb Sounder (TALIS) of China", Wang et al. propose a millimeter/sub-millimeter limb sounder for middle atmosphere observations. Using similar spectral ranges than those chosen for AURA/MLS, this instrument could be one of the few instruments operating after 2020, and, hence, it could be an essential contribution for the monitoring of long term changes of the middle atmosphere. The paper is clearly written and the study well explained (there are minor English issues that can be fixed during the edition). The previous missions and studies are properly acknowledged. I believe the manuscript should be published in AMT but not as it is.

My main concern is that I don't see a plan to really realize this instrument, and the study itself is not enough for a publication. As the authors stated, the concept is not

new (similar to Aura/MLS) and only a preliminary estimation of the measurement performance is shown. So I will recommend the authors to improve the manuscript with more information on their plan to realize the mission and on the observation strategy. Also, the instrument field of view and the related parameters (antenna size and scan velocity) are not given in the manuscript. There are key parameters for the assessment of limb sounding performance. Moreover, the retrieval vertical resolution that can be derived from the width of the averaging kernel, should be included in the discussion of the measurement error. The vertical resolution is indeed an essential characteristic of the retrieval performance.

Here below I provide ideas that, I think, could improve the paper.

1) I understand that the realization of such mission is uncertain, and a lot of details are not decided yet. However, it would be nice to know how the authors plan to realize it: When and to who this mission could be proposed? Launch date? Which technology will be used and is it tested with ground based systems? Where the value of Tsys come from? ...

2) What is the size of the antenna? The retrieval vertical resolution should be included in the description of the retrieval performance. The measurement performance of the main products (temperature, H2O, O3) should be compared with those of Aura/MLS.

3) The spectral resolution is lower than MLS at the center of key lines such as O2 @119 GHz and H2O@183 GHz (0.2 MHz vs 2 MHz). This choice has a significant impact on the retrievals in the mesosphere. It should be discussed.

4) More details on the observation strategy could be provided (latitude coverage, local time, ...). They are improtant when discussing the measurement of radical such as ClO, NO2, NO, HOCl, HO2 which have strong diurnal variations (see for instance Khosravi et al.: Diurnal variation of stratospheric and lower mesospheric HOCl, ClO and HO2 at the equator: comparison of 1-D model calculations with measurements by satellite instruments, Atmos. Chem. Phys., https://doi.org/10.5194/acp-13-7587-2013,

2013.)

Specific comments:

Page 1, L25: "Terahertz limb ...", Most of the statements in this sentence are specific of limb sounding in general. I would rather write "Limb sounders ... in a wide altitude range. In the terahertz domain, the measurement performances are independent of the day-night cycle."

Page 1, L29: The reference "Ochiai et al., 2017" may not be the best one here since it describes a proposed mission (as indicated further in the paper). The potential for wind measurement has been demonstrated in the mesosphere with Aura/MLS (Wu et al., 2008) and in the stratosphere with JEM/SMILES (Baron et al., 2013).

[Baron et al.: Observation of horizontal winds in the middle-atmosphere between 30S and 55N during the northern winter 2009-2010, Atmos. Chem. Phys., 13, 6049–6064, https://doi.org/10.5194/acp-13-6049-2013, 2013] [Wu et al., Mesospheric doppler wind measurements from Aura Microwave Limb Sounder, Adv. Space Res., 42, 2008]

TALIS should be able to provide wind information between 40 and 60 km (low Tsys and more than 10 strong lines). Are the authors plan to investigate the line-of-sight wind retrievals? If yes, this point could be added in the conclusion as future studies.

Page 2, L15 and 20: To my knowledge TLS is not a selected mission yet.

Page 3, Sect 2.1: What is the antenna size? What is the scan velocity?

Page 3, L18. The $O_2$ line will be measured with less bandwidth than Aura/MLS which is equipped filter bank at -/+ 4 GHz from the $O_2$ line center. This difference should be discussed as well as the consequences for temperature and pressure measurement in the upper-troposphere. Is the $O_2$ line at 239 GHz can compensate the loss of information compared to Aura/MLS?

Page 4, L21 "Using ... in the 643 GHz bands can measure..." should be "Using ... in

the 643 GHz band, one can measure. . .."

Page 4, L24 "Sato et al., 2013" should be "Sato et al., 2012".

Page 5, L5: ". . . because of the negative bias in ClO . . ." I find this statement unclear.

Page 5, Fig. 1-4: Stronger colors should be used. The linewidth could also be increased.

Page 7, Eq 1: This equation is not used in the study. I would remove it and discuss Eq 4 instead. Same for the sentence "The so-call phase function . . ." in P8L7.

Page 8, L6: The sentence "Thus the radiance can be converted . . ." is unclear. The so-call Brightness-temperature unit is obtained from the linear transformation of the intensity in SI using the Rayleigh-Jeans factor: see Eq 3 in Urban et al., 2004 (reference given in P8, L4).

Page 9,' L7: "The predicted radiance y_hat are compared . . . ". I disagree with this statement. The measurement is compared with a noiseless prediction as written in Eq9: chi2 depends on "y-F(x,b)" and not on "y-y_hat". This part should be rephrased.

Page 9, L15: The notation for the system temperature should be consistent with that used in Tab. 1 (Trec vs Tsys). More information/references on how the Tsys values are defined should be provided.

Page 9, L25: What the authors mean by "strict approach"?

Page 10, L22. Is "vertical separation" the spacing between retrieval altitudes? What is the instrument field-of-view? What is the altitude range between 2 spectra? Are they the same as in the Livesey and Snyder study?

Page 10, L30: The sentence "The true profiles are perturbed to be the a priori profiles" is not clear. Are the true profiles derived by perturbing a priori profiles?

Page 11, L1: I don't understand the factor sqrt(2). Should it be a factor of 2? Tsys for

[Figure]

SSB = 2 * Tsys for DSB with Eq6 definition (the sideband signals are divided by 2).

In Table 1, it should be clearly indicated that Tsys is for SSB. In Eq 10, it should be clearly indicated that there is a factor to be included for DSB case.

Page 11, L7: "...plotted in the figures." The figure numbers should be indicated "... plotted in Figs 5 to 21". The plots will be clearer if a grid would be added. Also a log scale should be used for the errors (difficult to see details if the errors are small).

Page 11, L15-19: The vertical resolution derived from the width of the averaging kernel should be described and discussed. This is a general comment that should be applied to all products. Also the results should be compared with those from Aura/MLS, at least for the most important ones (temperature, H2O and O3).

Page 11, L19: The sentence "Results of 643 GHz seems not very good" should be rephrased. For me the results look fine in the upper troposphere compared to the two other bands. The error is 2 K, like that for the 240 GHz band and the vertical resolution looks similar.

I think the 2 following information should be given: - Is the tangent height pressure retrieved? - The Zeeman effect polarizes and changes the shape of the O2 lines (Schwartz et al., 2006). Future studies will investigate the impacts in the measurement performance above 60 km. [Schwartz et al., EOS MLS forward model polarized radiative transfer for Zeeman-split oxygen lines, IEEE Transactions on Geoscience and Remote Sensing, doi:10.1109/TGRS.2005.862267]

Page 13, L3: I would rephrase the first sentence as "The H2O profile, another key parameter, can be measured ...."

Page 15, L2: "covering" should be "covers"

Page 15, L4: "The other two retrievals ... poor precision", I think this sentence is too negative since a good performance is achieved between 20 and 60 km with the 190 and 650 GHz bands.

Page 17, L1: "widely exist" could be removed (-> HNO3 is a common species in the stratosphere . . ..)

Page 17, L1: It is unclear what "stable lines" mean.

Page 17, L17: The statement "seems bad" is negative and not quantitative. I think it should be stated that the 643 GHz signal is stronger than that in the 240 GHz band, but it is strongly absorbed by O3 below about 30 km. However after averaging the measurements, information can be retrieved between 25 and 70 km with a precision better than 50%.

Page 19, L3: "can be retrieved by 190 GHz . . ." should be "can be retrieved from the band at 190 GHz . . .. while the band at 643 GHz . . ."

Page 21, L1-2: The statement "is not desirable" is not appropriate. This line will be "desirable" if the radiometer at 640 GHz is not working. Instead, I would state that the best retrievals are perform from the band at 640 GHz but information can also be retrieved from the radiometer at 190 GHz with less precision."

Page 23, Plot b: What happens at 32 and 65 km? There are strong increases of sensitivity.

Page 25, L4: "exist" should be removed (-> "There are several weak lines in the spectral. . .")

Page 28-29, The discussion for NO2 and NO should include the effects of their diurnal variation (this is also true for HO2, ClO). Around 40 km, NO2 vanishes in daytime and NO in nighttime. Since the apriori error is assumed proportional to the VMR, no good measurement sensitivity near 40 km can be found for NO2 in daytime and for NO nighttime. Therefore, I think it is important to clearly define which conditions are considered in the calculation and to discuss the consequences in term of retrieval sensitivity.

Page 33: The expression such as "... temperature < 2 K" should be corrected (e.g., ...

temperature with a precision < 2 K). Several similar cases occur in the conclusion.

Page 33, L4: Any plan to derive IWC? If yes, this could be indicated as future works.

Page 33, L17 The sentence "Although the single scan precision seems not very", as I already mentioned before, such statement is not appropriate. I think a statement such as "The best sensitivity is found between 70—90 km where the precision is better than . . ." would be better.

---

## Author Response (AR1)

**Reviewer 1**

We would like to sincerely thank the reviewer for his comments. We believe they help us to improve the manuscript significantly and give us many useful suggestions to improve the mission. We have corrected the manuscript according

5   reviewer's comments and answer the reviewer's question point by point below.

Reviewer comments are in italic blue, the manuscript modifications are in red. The answer "Done" means that the manuscript has been modified following exactly the reviewer comment.

10   **Reply to general comments**

*1) Where the two instruments differ is in the spectrometers used. EOS MLS has oldfashioned filter banks, whereas TALIS will use FFT spectrometers with many more channels. For this paper to demonstrate any novelty, it needs to show how the coverage of the TALIS spectrometers differs from that of the MLS filter banks, and to demonstrate the extent to which the improved coverage leads to improved quality of retrieval products. This would involve simulating retrievals with the two*

15   *sorts of spectrometers, with all other factors kept identical.*

Reply: We agree that spectrometer is the major difference between the two instruments. Figure 3 has been added to show the coverage of the TALIS spectrometers differs from that of the MLS filter banks. (see specific comments)

20   Section 4.2 is added to show the performance comparison between TALIS FFT spectrometer and MLS 'Standard' 25-channel spectrometer (Fig.9 to 11):

"As discussed in section 2.2, TALIS has similar bands to EOS MLS. The major difference between these two instruments is the spectrometers used in limb sounding. A simulation is performed to compare the performance of the main products between TALIS FFT spectrometer and EOS MLS 'Standard' 25-channel spectrometer. Figure 9 to 11 show the retrieval

25   products of TALIS and MLS, all the factors are identical except the spectrometer.

According to the simulation results, TALIS can do a better job than EOS MLS because of the wider bandwidth and finer resolution. Temperature precision of TALIS is improved 1–2 K compared with EOS MLS and the vertical resolution is improved about 2 km. $H_2O$ precision is improved about 2–10 %. $O_3$ precision is improved about 2–20 % and retrieved well in the mesosphere".

[Figure]

**Figure 9. Temperature product comparison between TALIS FFT spectrometer and MLS 'Standard' spectrometer using 118.75 GHz line. All other factors are identical.**

[Figure]

**Figure 10. H₂O product comparison between TALIS FFT spectrometer and MLS 'Standard' spectrometer using 183.31 GHz line. All other factors are identical.**

[Figure]

**Figure 11. O$_3$ product comparison between TALIS FFT spectrometer and MLS 'Standard' spectrometer using 235.71 GHz line. All other factors are identical.**

*2) The simulation reported has a number of failings. The most serious of these are the failure to properly consider the vertical resolution of the instrument, and the failure to describe the antenna characteristics assumed; these characteristics are the main factor limiting the vertical resolution of the instrument. For the paper to be published, the antenna characteristics should be described, and the inevitable tradeoff between resolution and precision considered properly. Another problem is the failure to address how information on the geometrical tangent height is to be incorporated into the retrieval of temperature and tangent pressure.*

Reply: We are sorry to omit these characteristics. Section 2.1 has been rephrased, more details about TALIS are added:

"The TALIS payload (Fig. 1) and its proposed scan characteristics are summarized in Table 1. The instrument will be set at a sun-synchronous orbit at a normal altitude of 600 km. The offset parabolic antenna is made of a single reflector with 1.6 m projective aperture and four independent feeds. The layout of four discrete feeds is shown in Fig. 2. Compared with the quasi-optical separation layout (such as MLS), this strategy is easier and has higher observation precision. But it will lead to an observed difference of about 20 km between the four radiometers. The widths of the field of view (FOV) at the tangent point are about 5.5, 3.8, 3.3, and 0.96 km respectively. The two-point calibration method is adopted by TALIS, and an extra calibration target is set at the bottom of the antenna. The extra target can be used to improve the calibration precision and evaluate the antenna effect and nonlinearity. At the beginning of the scan, TALIS will view the hot target (ambient temperature) and the extra target (lower temperature) in 3 s, and then it will scan the limb from 0 to 100 km vertically and obtain the spectra every 1 km with an integration time of 0.1 s, finally, it views the cold space at 200 km in 5 s. The process of retrace is the same (also record data) and giving a total period of about 36 s".

[Figure]

Figure 1. The schematic diagram of TALIS payload

Figure 2. The layout of the four antenna feeds

Table 1. Characteristics of the TALIS payload

| Satellite altitude | 600 km |
|---|---|
| Scan altitude | 0–100 km |
| LOS nadir angle | 66.07–68.17° (2.1°) |
| Scan velocity | 0.21° s$^{-1}$ (36 s scan$^{-1}$) |
| Spectrum integration time | 0.1 s (1 km ) |
| Antenna size | 1.6 m |
| Antenna vertical FOV | 5.5, 3.8, 3.3, 0.96 km |
| Spectrometer Bandwidth | 2 GHz |
| Spectrometer resolution | 2 MHz |
| LO frequency | 120, 190.1, 239.66, 642.87 GHz |

5    The proposed FOV of TALIS are 5.5, 3.8, 3.3, 0.96 km, and TALIS record the spectra every 1 km. According to the MLS study, there is a trade-off between the vertical resolution and information gained, and little information is lost by going from twelve to six surfaces per decade. Thus, we use 2.5 km in our simulation for simplicity since the FOV of TALIS are similar to MLS. We will discuss the best vertical resolution of TALIS in the future works.

10    The sentence in section 4.1 has been rephrased as follows (P10, L21):

"In this simulation, the scan altitude range is from 10 to 95 km and the spectra are obtained every 2.5 km. As the effective FOV of TALIS are similar to MLS, we use 2.5 km as the vertical resolution. It is the trade-off between the step of efficient limb observation and the optimum information can be obtained (Livesey and Snyder, 2004)".

The pressure and geometric height are linked by the hydrostatic equilibrium equation. The hydrostatic equilibrium equation needs the correct geometric height information of tangent point. The geometric height information is obtained by the satellite navigation system and the antenna view angle. It is usually different between the tangent point altitude as determined by geometry and reported by the instrument, so it should be retrieved as a state vector at first.

A statement has been added to temperature retrieval as follows (P11, L19):

"Once the temperature profile is retrieved, the pressure profile can be calculated from the hydrostatic equilibrium equation using a known pressure and temperature at a reference tangent point. The pressure profile is not a direct product and is not shown here".

**Reply to specific comments**

*Page 1 line 18: In "high single scan retrieval precision of 1 – 50%", remove the word "high". 1% can be considered high precision in this context, but 50% is not high precision in any context. I suppose you could replace "high" with "relatively high".*

Reply: Done. The sentence has been rephrased as follows:

"Chemical species such as $H_2O$, $O_3$, HCl show relatively high single scan retrieval precision of 1–20% over most of the useful range and ClO, $N_2O$, $HNO_3$ can be retrieved with a precision < 50%".

*Page 1 line 25–27: It seems odd to talk of this instrument as a "Terahertz limb sounder" when its highest frequency is 0.6 THz and its lowest frequency is below 0.2 THz.*

Reply: Yes, it is not very appropriate but it is named by our research project.

*Page 3 lines 5–7: The double sideband nature of both UARS MLS and Aura MLS has been a considerable nuisance to the instrument team, especially when attempting to obtain results in the important 300 hPa – 60 hPa range. It was a technological limitation at the time EOS MLS was built, which would have been too expensive to work around. Modern mm-wave technology includes sideband separating receivers. As TALIS appears still to be in the design phase, the authors might like to consider whether this technology would be appropriate.*

Reply: We thank the reviewer for this suggestion. Sideband separating receiver has been considered and we believe it is very useful in limb sounding. However, TALIS is limited to the technology and fund, it is difficult to use this receiver. Sideband separating receiver may be adopted in the future mission.

*Page 3 line 14: "TALIS covers most spectral bands of EOS MLS and extends them." It would perhaps be worth adding some sort of diagram with the EOS MLS and TALIS spectral bands overlaid on each other, so that the reader can quickly see by how much the TALIS coverage extends that of EOS MLS. The should also be a statement that EOS MLS had a band at 2.4 THz which TALIS will lack — there is no need of a diagram to show this.*

Reply: Figure 3 has been added to show the difference between MLS and TALIS spectral bands.

[Figure]

**Figure 3. Spectral bands of EOS MLS and TALIS**

The sentence has been rephrased as follows:

"TALIS covers most spectral bands of EOS MLS and extends them (see Fig 3), but lack the 2.4 THz band".

*Page 3 line 15: "... and lower noise of TALIS". The authors should probably state the Tsys values of the EOS MLS bands so*

15 *that the reader can easily see how much lower the noise level of TALIS will be. A cross-check against Waters et al. (2006) (see Table 1) indicates that the TALIS Tsys values are either effectively the same as those of EOS MLS or are about 30%–50% better. This is unlikely to be sufficient to allow easy measurement of a species which EOS MLS can not measure. If TALIS*

*can really do a better job than EOS MLS, it is likely to be on account of the bandwidth and resolution of the spectrometers rather than because of the slightly better measurement noise.*

Reply: We changed the Table 2 so that the Tsys value of TALIS and EOS MLS can be distinguished. The "lower noise" in this sentence has been removed.

The performance comparison between TALIS FFT spectrometer and MLS 'Standard' 25-channel spectrometer shows that TALIS can do a better job than EOS MLS because of the wider bandwidth and finer resolution.

*Page 4 Line 10: It is worth pointing out that the only SO$_2$ which is observable by instruments like EOS MLS and TALIS comes from volcanic eruptions.*

Reply: The sentence has been rephrased as follows:

"There is no obvious SO$_2$ emission with the standard profile present in the passband of 240 GHz radiometer. The only SO$_2$ which is observable by MLS comes from volcanic eruptions".

*Page 4 Lines 12–13: ". . .MLS demonstrated that SO$_2$ can be measured by 240 GHz radiometer cooperated with 190 GHz radiometer . . . ". What I actually said in Pumphrey et al. (2015) is that MLS measures SO$_2$ from three radiometers: 190 GHz, 240GHz and 640 GHz. The radiometers are not combined with each other. Rather, a separate SO$_2$ product is produced from each radiometer. Only the 240 GHz SO$_2$ product is recommended for general use.*

Reply: The sentence has been rephrased as follows:

"MLS demonstrated that SO$_2$ can be measured by 190 GHz, 240GHz, and 640 GHz radiometer, but only 240 GHz SO$_2$ product is recommended for general use".

*Page 4 line 14: "NO$_2$ is a unique species not covered by EOS MLS". This is entirely true. The authors should perhaps explain whether TALIS's ability to measure this species is due to improved bandwidth, resolution or radiometer noise. Figure 3 suggests that the measurement will be very difficult.*

Reply: TALIS's bandwidth and resolution are enough to detect the NO$_2$. However, the intensity of NO$_2$ emission is very weak, as the simulation result (Fig. 19) shows that only averaged measurements can be used to retrieve NO$_2$ profile.

The sentence has been rephrased as follows:

"NO$_2$ is a unique species not covered by EOS MLS, and TALIS's wider bandwidth and finer resolution have the potential ability to measure it".

*Page 8 line 5: This formula defines the Planck brightness temperature. It is not uncommon in microwave remote sensing*
5 *(especially in limb sounding) to work with the Rayleigh-Jeans brightness temperature because it is proportional to the radiance. The authors should be clear which brightness temperature they intend to use.*

Reply: We agree that it is common to work with the Rayleigh-Jeans brightness temperature in limb sounding. However, Planck's radiation law is the standard formula transform radiance to blackbody's absolute temperature. Rayleigh-Jeans law is
10 the approximation to Planck's law and its linear transformation will make the calculation simpler. But as the reference said, Rayleigh-Jeans law will lead to a brightness temperature error which varies with frequency and temperature. Thus, we want to use Planck function since it is widely used in the meteorological satellites all over the world such as Advanced Microwave Technology Sounder (ATMS) and FY-3 Microwave Humidity and Temperature Sounder (MWHTS). We will study the impact of these two different formulas in the future works.

Reference:

Weng, Fuzhong and Zou, Xiaolei.: Errors from Rayleigh–Jeans approximation in satellite microwave radiometer calibration systems, Applied Optics, 52, 505–508, https://doi.org/10.1364/AO.52.000505, 2013

20 We added a statement at the last of section 3.1 as follows:

"The measured radiance is transformed to brightness temperatures using the Planck's function".

*Page 10 section 4.1: This section omits a number of important items. Firstly, it does not state what antenna pattern is assumed when applying equation (5). This is important because the antenna pattern is the main limitation on vertical*
25 *resolution. The pattern does not always vary with frequency in the way one would naively expect. The 640GHz band of EOS MLS has better vertical resolution than the 190 GHz band, but nowhere near the three times better that you would expect from Rayleigh's criterion. This is because the 640GHz radiometer does not view the full aperture of the antenna. Secondly, nothing is said about the specific problem of temperature/pressure retrieval. We usually have two sources of information about where the antenna is pointing: pressure information that comes from the radiance measurements, and geometric*
30 *height information which comes from the satellite navigation system and the antenna view angle. These two pieces of information are linked by the hydrostatic equation, which itself depends on the temperature profile. For the temperature retrieval precisions presented to be credible, the reader needs to know how the geometric tangent height information was incorporated, and how precise it was assumed to be.*

Reply: We are sorry to omit these items and they are now added. The simulation antenna patterns have been added in Figure 8. The instrument field of view (FOV) are about 5.5, 3.8, 3.3, and 0.96 km respectively. Actually, we don't consider the antenna pattern in performance simulation because of the large computation time.

A statement has been added as follows:

"The simulation antenna patterns of the four radiometers are shown in Fig. 8. As the antenna calibration can be done by a linear function, it has no impact on the following simulation, so antenna pattern is not added in simulation below".

[Figure]

**Figure 8. The antenna patterns of TALIS**

The pressure and geometric height are linked by the hydrostatic equilibrium equation. The hydrostatic equilibrium equation needs the correct geometric height information of tangent point. The geometric height information is obtained by the satellite navigation system and the antenna view angle. It is usually different between the tangent point altitude as determined by geometry and reported by the instrument, so it should be retrieved as a state vector at first.

A statement has been added to temperature retrieval as follows (P11, L19):

"Once the temperature profile is retrieved, the pressure profile can be calculated from the hydrostatic equilibrium equation using a known pressure and temperature at a reference tangent point. The pressure profile is not a direct product and is not shown here".

*Page 14, figures 6–10:* The precision becomes very much poorer very suddenly at a height of 25 km. The authors should explain why this is. The figures make it appear as if the retrieval grid changes vertical resolution at this point from a grid which is coarser than the achievable vertical resolution to one which is finer than the achievable vertical resolution. This will inevitably make the retrieval precision appear far worse below 25 km, but this does not mean that the performance of
the instrument itself is far worse below 25 km. The authors should perhaps try showing averaging kernels calculated for a far finer grid, so that the true vertical resolution of the instrument can be assessed. They could then choose a retrieval grid which matches the achievable resolution better.

Reply: We agree that the possible explanation for the poor precision below 25 km might be the too finer grid resolution. The
10    retrieval grid resolution used in simulation is 1 km below 25 km, 2.5 km below 50 km and 5 km above 50 km. The best retrieval grid resolution needs to be further studied.

A statement has been added in conclusions as follows:

"As the results show that the precision of some products become much poorer suddenly at a height of 25 km, it seems that
15    the retrieval grid resolution does not match the achievable resolution".

*Page 23 figure 12:* The mixing ratio of CO changes over a very large range. In my experience, the only way to show a vertical profile over the 10 km–90 km range is to use a logarithmic mixing ratio scale.

20    Reply: Done.

**Technical corrections**

25    Reply: Done.

Reply: Done.

Reply: Done.

*Page 5, figures 1 – 4: The different lines on these figures can be quite hard to distinguish and to match up with the legend. It would help if the authors were to make the lines slightly thicker and to ensure that they choose strongly-contrasting colours.*

5 *(They should continue to avoid pure yellow (#ffff00) and pure green (#00ff00) as these colours can be hard to see on a white ground.) The vertical scale currently goes from 1K to 1000 K, but the data do not cover this entire range. If the software used will permit, the vertical scale should be reduced to cover 2 K to 300K*

Reply: Done. All figures have been plotted again.

*Page 4 line 16: "as more spectral" should be "as many spectral"*

Reply: Done.

15 *Page 4 lines 24–25 and throughout the paper: in LATEX, use a non-breaking thinspace (\,) between a number and its unit in order to avoid a line break at that point. Here, write 635.87\,GHz.*

Reply: Done.

20 *Page 8 line 3: T should be in math mode so that it comes out in math italic (T). In LATEX, write $T$, not T.*

Reply: Done.

*Diaeresis out of position over de la Nöe's name. Also, "la" does not have a capital letter.*

Reply: Done.

**Reviewer 2**

We would like to sincerely thank the reviewer for his comments. We believe they help us to improve the manuscript significantly and give us many useful suggestions to improve the mission. We have corrected the manuscript according

5   reviewer's comments and answer the reviewer's question point by point below.

Reviewer comments are in italic blue, the manuscript modifications are in red. The answer "Done" means that the manuscript has been modified following exactly the reviewer comment.

10  **Reply to general comments**

*1) I understand that the realization of such mission is uncertain, and a lot of details are not decided yet. However, it would be nice to know how the authors plan to realize it: When and to who this mission could be proposed? Launch date? Which technology will be used and is it tested with ground based systems? Where the value of Tsys come from? ...*

15  Reply: We appreciate your understanding, and a lot of details are not finally decided at the time this manuscript was written. More details about our plan have been added to the manuscript now.

The information about TALIS mission has been added in section 1:

"THz Atmospheric Limb Sounder (TALIS) is the pre-research project of civil aerospace technology proposed by China

20  National Space Administration (CNSA) ... The pre-research will be completed in 2020 and a prototype will be tested. The satellite mission equipped with TALIS will be proposed around 2021".

Section 2.1 has been rephrased, more details about the instrument have been added:

"The TALIS payload (Fig. 1) and its proposed scan characteristics are summarized in Table 1. The instrument will be set at a

25  sun-synchronous orbit at a normal altitude of 600 km. The offset parabolic antenna is made of a single reflector with 1.6 m projective aperture and four independent feeds. The layout of four discrete feeds is shown in Fig. 2. Compared with the quasi-optical separation layout (such as MLS), this strategy is easier and has higher observation precision. But it will lead to an observed difference of about 20 km between the four radiometers. The widths of the field of view (FOV) at the tangent point are about 5.5, 3.8, 3.3, and 0.96 km respectively. The two-point calibration method is adopted by TALIS, and an extra

30  calibration target is set at the bottom of the antenna. The extra target can be used to improve the calibration precision and evaluate the antenna effect and nonlinearity. At the beginning of the scan, TALIS will view the hot target (ambient temperature) and the extra target (lower temperature) in 3 s, and then it will scan the limb from 0 to 100 km vertically and

obtain the spectra every 1 km with an integration time of 0.1 s, finally, it views the cold space at 200 km in 5 s. The process of retrace is the same (also record data) and giving a total period of about 36 s".

[Figure]

[Figure]

**Figure 1. The schematic diagram of TALIS payload**

**Figure 2. The layout of the four antenna feeds**

**Table 1. Characteristics of the TALIS payload**

| Satellite altitude | 600 km |
|---|---|
| Scan altitude | 0–100 km |
| LOS nadir angle | 66.07–68.17° (2.1°) |
| Scan velocity | 0.21° $s^{-1}$ (36 s $scan^{-1}$) |
| Spectrum integration time | 0.1 s (1 km ) |
| Antenna size | 1.6 m |
| Antenna vertical FOV | 5.5, 3.8, 3.3, 0.96 km |
| Spectrometer Bandwidth | 2 GHz |
| Spectrometer resolution | 2 MHz |
| LO frequency | 120, 190.1, 239.66, 642.87 GHz |

*2) What is the size of the antenna? The retrieval vertical resolution should be included in the description of the retrieval performance. The measurement performance of the main products (temperature, $H_2O$, $O_3$) should be compared with those of Aura/MLS.*

Reply: The size of the antenna is 1.6 m. The retrieval vertical resolution has been added to each product discussion.

Section 4.2 is added to show the performance comparison between TALIS FFT spectrometer and MLS 'Standard' 25-channel spectrometer (Fig.9 to 11):

"As discussed in section 2.2, TALIS has similar bands to EOS MLS. The major difference between these two instruments is the spectrometers used in limb sounding. A simulation is performed to compare the performance of the main products between TALIS FFT spectrometer and EOS MLS 'Standard' 25-channel spectrometer. Figure 9 to 11 show the retrieval products of TALIS and MLS, all the factors are identical except the spectrometer.

5    According to the simulation results, TALIS can do a better job than EOS MLS because of the wider bandwidth and finer resolution. Temperature precision of TALIS is improved 1–2 K compared with EOS MLS and the vertical resolution is improved about 2 km. $H_2O$ precision is improved about 2–10 %. $O_3$ precision is improved about 2–20 % and retrieved well in the mesosphere".

[Figure]

10    **Figure 9. Temperature product comparison between TALIS FFT spectrometer and MLS 'Standard' spectrometer using 118.75 GHz line. All other factors are identical.**

[Figure]

**Figure 10. H₂O product comparison between TALIS FFT spectrometer and MLS 'Standard' spectrometer using 183.31 GHz line. All other factors are identical.**

[Figure]

**Figure 11. O₃ product comparison between TALIS FFT spectrometer and MLS 'Standard' spectrometer using 235.71 GHz line. All other factors are identical.**

*3) The spectral resolution is lower than MLS at the center of key lines such as O₂ @119 GHz and H₂O@183 GHz (0.2 MHz vs 2 MHz). This choice has a significant impact on the retrievals in the mesosphere. It should be discussed.*

Reply: 'Standard' 25-channel spectrometers of MLS have the poor spectral resolution. So Digital autocorrelator (DAC) spectrometers are needed to provide the finer spectral resolution for measurements in the mesosphere. However, according to the sensitivity formula, 0.2 MHz resolution will lead to a large noise of about 6-7 K, it needs a trade-off between resolution and noise. Our simulation results show that TALIS can measure O₂ and H₂O from upper-troposphere to mesosphere well.

*4) More details on the observation strategy could be provided (latitude coverage, local time, ...). They are improtant when discussing the measurement of radical such as ClO, NO₂, NO, HOCl, HO₂ which have strong diurnal variations (see for instance Khosravi et al.: Diurnal variation of stratospheric and lower mesospheric HOCl, ClO and HO₂ at the equator: comparison of 1-D model calculations with measurements by satellite instruments, Atmos. Chem. Phys., https://doi.org/10.5194/acp-13-7587-2013 2013.)*

Reply: We agree that the observation strategy is very important. TALIS is designed to be global coverage, and the local time is not determined yet since it is the pre-research. Thank you for your comments. We will consider the observation strategy with the satellite mission.

**Reply to specific comments**

*Page 1, L25: "Terahertz limb ...", Most of the statements in this sentence are specific of limb sounding in general. I would rather write "Limb sounders ... in a wide altitude range. In the terahertz domain, the measurement performances are independent of the day-night cycle."*

Reply: Done.

*Page 1, L29: The reference "Ochiai et al., 2017" may not be the best one here since it describes a proposed mission (as indicated further in the paper). The potential for wind measurement has been demonstrated in the mesosphere with Aura/MLS (Wu et al., 2008) and in the stratosphere with JEM/SMILES (Baron et al., 2013).*
*TALIS should be able to provide wind information between 40 and 60 km (low Tsys and more than 10 strong lines). Are the authors plan to investigate the line-of-sight wind retrievals? If yes, this point could be added in the conclusion as future studies.*

Reply: Done. The reference has been changed.

Indeed, wind measurement has been considered in our mission and whether the 655 GHz spectrometer should be included is also being discussed. This will be discussed in the future works.

This point has been added in the conclusion as follows:
"Apart from these products, some potential products will be discussed in the future works. Line-of-sight wind is important information which could be measured by TALIS. Cloud IceWater Content (IWC) is also an essential product provided by passive microwave radiometer. Future studies will also investigate the Zeeman effect as it polarizes and changes the shape of the $O_2$ lines".

*Page 2, L15 and 20: To my knowledge TLS is not a selected mission yet.*

Reply: Done. The sentence has been removed.

*Page 3, Sect 2.1: What is the antenna size? What is the scan velocity?*

Reply: The antenna size is 1.6 m, and the scan velocity is $0.21°s^{-1}$.

*Page 3, L18. The $O_2$ line will be measured with less bandwidth than Aura/MLS which is equipped filter bank at -/+ 4 GHz from the $O_2$ line center. This difference should be discussed as well as the consequences for temperature and pressure measurement in the upper-troposphere. Is the $O_2$ line at 239 GHz can compensate the loss of information compared to Aura/MLS?*

Reply: The 'Wide' filters of MLS make measurements extending down into the troposphere. Compared with MLS, the 118 GHz temperature product of TALIS will lack some information in the upper-troposphere. However, our simulation shows that the 240 GHz temperature product can provide the information in the upper-troposphere.

A statement has been added to temperature retrieval as follows (P11, L18):

"'Wide' filters of MLS make measurements extending down into the troposphere, and TALIS is lack of the information. However, the retrieval precision of 240 GHz band is better in the upper troposphere and the 240 GHz product can compensate for the loss of information. Result of 643 GHz band looks similar to that of 240 GHz band".

*Page 4, L21 "Using ... in the 643 GHz bands can measure ... " should be "Using ... in the 643 GHz band, one can measure ... ."*

Reply: Done.

Reply: Done.

*Page 5, L5: "... because of the negative bias in ClO ..." I find this statement unclear.*

Reply: We are sorry that this is a wrong statement and has been removed.

*Page 5, Fig. 1-4: Stronger colors should be used. The linewidth could also be increased.*

Reply: Done. All figures have been plotted again.

*Page 7, Eq 1: This equation is not used in the study. I would remove it and discuss Eq 4 instead. Same for the sentence "The so-call phase function ..." in P8L7.*

Reply: Done. The paragraph has been rephrased as follows:

"Radiative transfer describes the emission, propagation, scattering, and absorption of electromagnetic radiation (Mätzler., 2006). Scattering usually can be neglected above the upper troposphere because of the cloud free. In this way and assuming Local Thermodynamic Equilibrium (LTE), the formal solution of the radiative transfer equation is defined by

$$I_v(S_2) = I_v(S_1)e^{-\tau_{v(S_1,S_2)}} + \int_{S_1}^{S_2} \alpha_v(s)B_v(T)e^{-\tau_{v(S,S_2)}}ds, \tag{1}$$

where $I_v$ is the radiance at frequency $v$ reaching the sensor, $\alpha$ is the absorption coefficient and $\tau$ is the opacity or optical thickness. $B_v$ stands for the atmospheric emission which is given by Planck function describing the radiation of a black-body at temperature $T$ and frequency $v$ per unit solid angle, unit frequency interval and unit emitting surface (Urban et al., 2004):

$$B_v(T) = \frac{2hv^3}{c^2} \frac{1}{e^{hv/k_BT}-1}, \tag{2}$$

where $h$ is the Planck constant, $c$ is the speed of light, $k_B$ denotes Boltzmann constant".

*Page 8, L6: The sentence "Thus the radiance can be converted ..." is unclear. The so-call Brightness-temperature unit is obtained from the linear transformation of the intensity in SI using the Rayleigh-Jeans factor: see Eq 3 in Urban et al., 2004 (reference given in P8, L4).*

Reply: As discussed in reference, the so-call Brightness-temperature can be obtained by Planck's law or Rayleigh-Jeans law. Actually, Planck's radiation law is the standard formula transform radiance to blackbody's absolute temperature. Rayleigh-Jeans law is the approximation to Planck's law and its linear transformation will make the calculation simpler. However, Rayleigh-Jeans law will lead to a brightness temperature error which varies with frequency and temperature. Although Rayleigh-Jeans law is commonly used in limb sounding, we want to use Planck function since it is widely used in the meteorological satellites all over the world such as Advanced Microwave Technology Sounder (ATMS) and FY-3D Microwave Humidity and Temperature Sounder (MWHTS). We will study the impact of these two different formulas in further works.

Reference:

Weng, Fuzhong and Zou, Xiaolei.: Errors from Rayleigh–Jeans approximation in satellite microwave radiometer calibration systems, Applied Optics, 52, 505–508, https://doi.org/10.1364/AO.52.000505, 2013

The sentence has been removed and we added a statement at the last of section 3.1 as follows:

"The measured radiance is transformed to brightness temperatures using the Planck's function".

*Page 9, L7: "The predicted radiance y_hat are compared ... ". I disagree with this statement. The measurement is compared with a noiseless prediction as written in Eq9: chi2 depends on "y-F(x,b)" and not on "y-y_hat". This part should be rephrased.*

Reply: We are very sorry for our carelessness. The sentence has been rephrased as follows:

"The noiseless predicted radiance $F(x, b)$ are compared with the observed radiance $y$ so that the unknow state which minimize the cost function $\chi^2$ could be found".

*Page 9, L15: The notation for the system temperature should be consistent with that used in Tab. 1 (Trec vs Tsys). More information/references on how the Tsys values are defined should be provided.*

Reply: Done. All the Trec have been replaced by Tsys. System noise temperature (Tsys) means the sum of receiver noise temperature (Trec) and the atmosphere temperature received by the antenna ($T_A$). It is an equivalent temperature determined by receivers.

*Page 9, L25: What the authors mean by "strict approach"?*

Reply: "strict approach" means OEM can describe the error completely, the sentence has been rephrased as follows:

"The OEM method provides an approach to describe the retrieval error completely".

*Page 10, L22. Is "vertical separation" the spacing between retrieval altitudes? What is the instrument field-of-view? What is the altitude range between 2 spectra? Are they the same as in the Livesey and Snyder study?*

Reply: "vertical separation" means the altitude space between two spectra assumed in our simulation. The simulation antenna patterns of the four radiometers are shown in Fig. 8. The proposed FOV of TALIS are 5.5, 3.8, 3.3, 0.96 km, and TALIS record the spectra every 1 km. According to the study of Livesey and Snyder, there is a trade-off between the vertical resolution and information gained, and little information is lost by going from twelve to six surfaces per decade. Thus, we use 2.5 km in our simulation for simplicity since the FOV of TALIS are similar to MLS. We will discuss the best vertical resolution of TALIS in the future works.

The sentence has been rephrased as follows:

"In this simulation, the scan altitude range is from 10 to 95 km and the spectra are obtained every 2.5 km. As the effective FOV of TALIS are similar to MLS, we use 2.5 km as the vertical resolution. It is the trade-off between the step of efficient limb observation and the optimum information can be obtained (Livesey and Snyder, 2004)".

5 *Page 10, L30: The sentence "The true profiles are perturbed to be the a priori profiles" is not clear. Are the true profiles derived by perturbing a priori profiles?*

Reply: The a priori profiles are derived by perturbing true profiles. The sentence has been rephrased as follows:
"The true species profiles are multiplied by a factor of 1.1 to be the a priori profiles, and the true temperature profile is added
10 a 5 K offset to be the a priori profile".

*Page 11, L1: I don't understand the factor sqrt(2). Should it be a factor of 2? Tsys for SSB = 2 \* Tsys for DSB with Eq6 definition (the sideband signals are divided by 2). In Table 1, it should be clearly indicated that Tsys is for SSB. In Eq 10, it should be clearly indicated that there is a factor to be included for DSB case.*

Reply: The factor of sqrt(2) comes from the noise equivalent bandwidth. As described in the reference: "It should be stated at this stage that the radiometer bandwidth $B$ in the sensitivity formula is the predetection bandwidth, that is, the IF bandwidth, and not the total RF bandwidth, which is $2B$ in DSB receivers".

20 Reference:
Skou N, Vine D L.: Microwave Radiometer Systems: Design and Analysis, Norwood Ma Artech House P, 2006, 26-27

Tsys is the system noise temperature for the band. In Table 1, this is a single-sideband value for 118 GHz radiometer, and double-sideband value for other radiometers. A statement is added to Table 1.

The sentence has been rephrased as follows:
"where $T_{sys}$ is the system noise temperature which is the sum of receiver noise temperature and the atmosphere temperature received by the antenna, $\beta$ is the noise equivalent bandwidth and $d\tau$ is the integration time for measuring a single spectrum. When it comes to DSB radiometer, the $\epsilon$ need to be divided by $\sqrt{2}$".

*Page 11, L7: "... plotted in the figures." The figure numbers should be indicated "... plotted in Figs 5 to 21". The plots will be clearer if a grid would be added. Also a log scale should be used for the errors (difficult to see details if the errors are small).*

Reply: Done. All figures have been plotted again.

*Page 11, L15-19: The vertical resolution derived from the width of the averaging kernel should be described and discussed.*
5  *This is a general comment that should be applied to all products. Also the results should be compared with those from Aura/MLS, at least for the most important ones (temperature, H₂O and O₃).*

Reply: The vertical resolution has been added to all products.
The comparison of TALIS and MLS retrievals are shown in section 4.2. (see general comments)

*Page 11, L19: The sentence "Results of 643 GHz seems not very good" should be rephrased. For me the results look fine in the upper troposphere compared to the two other bands. The error is 2 K, like that for the 240 GHz band and the vertical resolution looks similar.*
*I think the 2 following information should be given: - Is the tangent height pressure retrieved? - The Zeeman effect polarizes*
15  *and changes the shape of the O2 lines (Schwartz et al., 2006). Future studies will investigate the impacts in the measurement performance above 60 km. [Schwartz et al., EOS MLS forward model polarized radiative transfer for Zeeman-split oxygen lines, IEEE Transactions on Geoscience and Remote Sensing, doi:10.1109/TGRS.2005.862267]*

Reply: The sentence has been rephrased as follows:
20  "Result of 643 GHz looks similar in the upper troposphere compared to the 240 GHz band".

Tangent height pressure can be derived from the hydrostatic equilibrium equation, it is not retrieved in this simulation. A statement has been added as follows:
"Once the temperature profile is retrieved, the pressure profile can be calculated from the hydrostatic equilibrium equation
25  using a known pressure and temperature at a reference tangent point. The pressure profile is not a direct product and is not shown here".

Zeeman effect is not considered because of the large computational complexity. These impacts will be concerned in the future works. It has been added in the conclusion.

*Page 13, L3: I would rephrase the first sentence as "The H₂O profile, another key parameter, can be measured ..."*

Reply: Done.

*Page 15, L2: "covering" should be "covers"*

Reply: Done.

*Page 15, L4: "The other two retrievals ... poor precision", I think this sentence is too negative since a good performance is achieved between 20 and 60 km with the 190 and 650 GHz bands.*

Reply: This sentence has been rephrased as follows:

10 "The other two bands also show good performance from 25 to 60 km with a single scan precision < 5%".

*Page 17, L1: "widely exist" could be removed (-> HNO₃ is a common species in the stratosphere ...)*

Reply: Done.

*Page 17, L1: It is unclear what "stable lines" mean.*

Reply: It has been rephrased as follows:
"relatively strong lines".

*Page 17, L17: The statement "seems bad" is negative and not quantitative. I think it should be stated that the 643 GHz signal is stronger than that in the 240 GHz band, but it is strongly absorbed by $O_3$ below about 30 km. However after averaging the measurements, information can be retrieved between 25 and 70 km with a precision better than 50%.*

25 Reply: This sentence has been rephrased as follows:
"The 643 GHz signal is stronger than that in the 240 GHz band, but it is strongly absorbed by $O_3$ below about 30 km. However, after averaging the measurements, information can be retrieved between 25 and 70 km with a precision better than 50%".

30 *Page 19, L3: "can be retrieved by 190 GHz ..." should be "can be retrieved from the band at 190 GHz .... while the band at 643 GHz ..."*

Reply: Done.

*Page 21, L1-2: The statement "is not desirable" is not appropriate. This line will be "desirable" if the radiometer at 640 GHz is not working. Instead, I would state that the best retrievals are perform from the band at 640 GHz but information can also be retrieved from the radiometer at 190 GHz with less precision."*

Reply: This sentence has been rephrased as follows:

"However, the result shows that the best retrievals are performed from the band at 643 GHz but information can also be retrieved from the 190 GHz radiometer with less precision".

10  *Page 23, Plot b: What happens at 32 and 65 km? There are strong increases of sensitivity.*

Reply: It is not clear what happened yet, maybe it is related to the grid resolution. We will study this problem in the future works.

15  *Page 25, L4: "exist" should be removed (-> "There are several weak lines in the spectral…")*

Reply: Done.

*Page 28-29, The discussion for $NO_2$ and NO should include the effects of their diurnal variation (this is also true for $HO_2$,*
20  *ClO). Around 40 km, $NO_2$ vanishes in daytime and NO in nighttime. Since the apriori error is assumed proportional to the VMR, no good measurement sensitivity near 40 km can be found for $NO_2$ in daytime and for NO nighttime. Therefore, I think it is important to clearly define which conditions are considered in the calculation and to discuss the consequences in term of retrieval sensitivity.*

25  Reply: In simulation, $NO_2$ is the nighttime profile and NO is the daytime profile. The conditions have been added.

*Page 33: The expression such as "… temperature < 2 K" should be Done. (e.g., … temperature with a precision < 2 K). Several similar cases occur in the conclusion.*

30  Reply: Done.

*Page 33, L4: Any plan to derive IWC? If yes, this could be indicated as future works.*

Reply: Yes. It has been added in the conclusion as future works.

*Page 33, L17 The sentence "Although the single scan precision seems not very", as I already mentioned before, such statement is not appropriate. I think a statement such as "The best sensitivity is found between 70~90 km where the precision*

5  *is better than …" would be better.*

Reply: The sentence has been rephrased as follows:

[revised manuscript text omitted]

---

## Referee Report (RR1)

The manuscript "Performance Evaluation of THz Atmospheric Limb Sounder (TALIS) of China" by Wang et al. has been significantly improved compared to the first version. I still have few minor comments before it could be published. Though the manuscript is clearly written, they are still few English mistakes that should be corrected.

I found inconsistency in some figures given in the text that should be checked (see list of comments).

They may be due to misunderstanding on my part.

Minor comments:

Abstract

A sentence on the comparison with AURA/MLS should be added in the abstract.

P2 L17: "other abundant chemical species" -> "other trace chemical species"

P3P8: "5.5, 3.8, 3.3, and 0.96 km, respectively." -> "5.5, 3.8, 3.3, and 0.96 km at 118, 190, 240, 640 GHz, respectively."

Table 1: Since there is an angle offset between the radiometers, how are defined the LOS angles?

I have trouble to understand the relationship "0.1 s integration time -> 1 km tangent height spacing".

If the total time for scanning from 0 to 100 km is 28 sec (36 - 5-3), 0.1 sec should be 0.35 km (100/28.*0.1). Do I misunderstand something?

P5L9-10: I would recommend the authors to add the MLS reference on the 118 GHz line processing since their analysis of this line is not complete (e.g., Zeeman effect). [Schwartz et al., EOS MLS forward model polarized radiative transfer for Zeeman-split oxygen, IEEE Transactions on Geoscience and Remote Sensing, doi:10.1109/TGRS.2005.862267]

P6: A reference with MLS HCl could be added. For example: Lary, D.J., O. Aulov, "Space-based measurements of HCl: Intercomparison and historical context", *Journal of Geophysical Research 113*, D15S04, doi:10.1029/2007JD008715, 2008

P11L8, The reference given by the authors in their answer to my first review (referee 2) should be given ("Skou N, Vine D L.: Microwave Radiometer Systems: Design and Analysis, Norwood Ma Artech House P, 2006, 26-27"). Also the fact that the noise equivalent bandwidth of a DSB receiver is twice that of a SSB receiver should also be indicated explicitly. For example: "When it comes to DSB radiometer, the $\epsilon$ need to be divided by sqrt(2) and \beta is twice that of the SSB. "

This equation gives the noise STD expressed in the unit of "brightness temperature" (Rayleigh Jeans temperature) but the authors explained in their answer to my first review that they do not use this unit. They should clarify the unit issue here.

P12, Eq16: I think the equation is not used and should be removed.

P12L9-11: The following section is not clear:

"The ideal rectangle channel response function is used. The simulation antenna patterns of the four radiometers are shown in Fig. 8. As the antenna calibration can be done by a linear function, it has no impact on the following simulation, so antenna pattern is not added in simulation below."

Do the authors mean that they use an ideal rectangle channel function instead of the antenna patterns in Fig 8 ? If yes, they should rephrase it and indicate the widths for all radiometer (one function per radiometer). I think that such an approximation is fine for the error estimations presented in this study, but it has to be clearly explained. Note that the scan vertical velocity should be taken into account.

What does "antenna calibration" mean in "As the antenna calibration can be done by a linear function, it has no impact on the following simulation, …"?

P12L15: It is stated that "… the spectra are obtained every 2.5 km." but in Tab. 1 it is shown 1 km. What is the correct value?

P12L15: Is "we use 2.5 km as the vertical resolution" for the retrieval layer? This statement is unclear. Is-it the tangent height spacing or antenna FOV vertical resolution or the retrieval layer resolution?

P12L7: "The retrieval grid resolution is 1 km below 25 km, 2.5 km below 50 km, and 5 km above 50 km."

The choice of 1 km for retrieval layer below 25 km should be explained. In my point of view it is not an "optimal" choice for 2 reasons: 1) it is not consistent with the 2.5 km spacing of the tangent heights, and 2) only the 640 GHz radiometer can measure with a resolution better than 1 km.

P13L8: I still have trouble with the noise estimation. If I applied Eq 9 to the DBS receiver and taking into account that beta is 2 x spectrometer bandwidth I have:

Noise STD at 640 GHz = 3.35 K (=3000/ sqrt(0.1*4.e6)/sqrt(2), with dt = 0.1 sec, beta=4.e6 Hz)

This value is smaller than that given by the authors (5 K) by a factor close to sqrt(2).

Also the radiometer should be indicated in the sentence:

"… to be 2 K, 1.7 K, 1.7 K, and 5 K , respectively." -> "… to be 2 K, 1.7 K, 1.7 K, and 5 K at 118, 190, 240, 640 GHz, respectively."

Fig9: The plots on the third panel are not visible. A log scale should be used.

P15L1: What is the error covariance matrix considered here: Sm (Eq 15) or Sm + Sn (Eq 15 and Eq 16)?

In several figures (e.g., Fig 13), the retrieval error is larger than 100%. How it is possible if the a priori error is set at 100%

P15L10: The altitude range for good retrievals should be indicated (15-85 km).

P15L12: It is difficult to check the vertical resolution between 15 and 25 km, though the value of 2.5 km looks correct above 20 km. Below 20 km the resolution looks poorer.

P15L13: "and TALIS is lack of the information." -> "where TALIS lacks sensitivity."

P15L15: The authors should provide values to support the statement. For example:

"is better in in the upper troposphere (error of 2 K for a vertical resolution of 3-4 km between 10 and 15 km)"

P17L7: It worth to indicate that O3 can be measure down to 10 km but with relatively low precision (50% and vertical resolution of 3-5 km).

P21L4-8: I think the diurnal change of ClO should be discussed. The ClO profile corresponds to daytime and the nighttime relative precision will be worse between 30-40 km (ClO vanishes during nighttime).

In the polar regions, the relative precision will be high between 20 and 25 km during chlorine activation.

---

## Author Response (AR2)

**Reviewer 2**

We would like to sincerely thank the reviewer for his comments. We believe they help us to improve the manuscript significantly and give us many useful suggestions to improve the mission. We have corrected the manuscript according reviewer's comments and answer the reviewer's question point by point below.

Reviewer comments are in italic blue, the manuscript modifications are in red. The answer "Done" means that the manuscript has been modified following exactly the reviewer comment.

**Reply to Minor comments**

*Abstract*

*A sentence on the comparison with AURA/MLS should be added in the abstract.*

Reply: Done. A sentence has been added:

"A theoretical simulation is performed to estimate the retrieval precision of the main targets and to compare them with that of Aura MLS standard spectrometers"

*P2 L17: "other abundant chemical species" -> "other trace chemical species"*

Reply: Done.

*P3P8: "5.5, 3.8, 3.3, and 0.96 km, respectively." -> "5.5, 3.8, 3.3, and 0.96 km at 118, 190, 240, 640 GHz, respectively."*

Reply: Done.

*Table 1: Since there is an angle offset between the radiometers, how are defined the LOS angles? I have trouble to understand the relationship "0.1 s integration time -> 1 km tangent height spacing". If the total time for scanning from 0 to 100 km is 28 sec (36 - 5-3), 0.1 sec should be 0.35 km (100/28.\*0.1). Do I misunderstand something?*

Reply: The LOS angle is the same, but there exists sequence between the radiometers. For example, 118 Ghz will scan the surface firstly, then 643 GHz will scan it, 190 GHz is the last. All the data will be recorded after 118 GHz start to work, so it has no impact on observation.

The total time of 36 second defined in paper includes a scan and retrace (retrace also records data, it will spend 18 second ), for one single scan it is 18 second, so the time used to scan from 0-100 km is 10 seconds.

*P5L9-10: I would recommend the authors to add the MLS reference on the 118 GHz line processing since their analysis of*

*this line is not complete (e.g., Zeeman effect). [Schwartz et al., EOS MLS forward model polarized radiative transfer for Zeeman-split oxygen, IEEE Transactions on Geoscience and Remote Sensing, doi:10.1109/TGRS.2005.862267]*

Reply: Done. A sentence has been added:

"In addition, the Zeeman effect will affect the $O_2$ line, the influence should be studied (Schwartz et al, 2006)"

*P6: A reference with MLS HCl could be added. For example: Lary, D.J., O. Aulov, "Space-based measurements of HCl: Intercomparison and historical context", Journal of Geophysical Research 113, D15S04, doi:10.1029/2007JD008715, 2008*

Reply: Done.

10   *P11L8, The reference given by the authors in their answer to my first review (referee 2) should be given ("Skou N, Vine D L.: Microwave Radiometer Systems: Design and Analysis, Norwood Ma Artech House P, 2006, 26-27"). Also the fact that the noise equivalent bandwidth of a DSB receiver is twice that of a SSB receiver should also be indicated explicitly. For example: "When it comes to DSB radiometer, the $\epsilon$ need to be divided by sqrt(2) and \beta is twice that of the SSB. "*

Reply: We are very sorry for our fault. We have checked that the noise equivalent bandwidth of a DSB receiver is the same

15   as an SSB receiver. It is our misunderstanding in former answer. The sentence has been deleted.

As the Reviewer 3 comments: "I'm not sure about the sqrt(2). Surely it is only needed if the Tsys is defined in terms of single sideband for a double sideband receiver."

Tsys in Table 1 has been updated in terms of our latest design and a new simulation is performed. Tsys is 1000 K, 1000 K, 1000 K, 2300 K for 118 GHz, 190 GHz, 240 GHz, 643 GHz radiometers. This is a single-sideband value for 118 GHz

20   radiometer, and double-sideband value for other radiometers. The corresponding noise is 2.2 K, 2.2 K, 2.2 K, 5.1 K.

*This equation gives the noise STD expressed in the unit of "brightness temperature" (Rayleigh Jeans temperature) but the authors explained in their answer to my first review that they do not use this unit. They should clarify the unit issue here*

Reply: Done. A sentence has been added:

25   "Planck function is used to compute the brightness temperature"

*P12, Eq16: I think the equation is not used and should be removed.*

Reply: Done.

30   *P12L9-11: The following section is not clear: "The ideal rectangle channel response function is used. The simulation antenna patterns of the four radiometers are shown in Fig. 8. As the antenna calibration can be done by a linear function, it has no impact on the following simulation, so antenna pattern is not added in simulation below." Do the authors mean that they use an ideal rectangle channel function instead of the antenna patterns in Fig 8 ? If yes, they should rephrase it and*

*indicate the widths for all radiometer (one function per radiometer). I think that such an approximation is fine for the error estimations presented in this study, but it has to be clearly explained. Note that the scan vertical velocity should be taken into account. What does "antenna calibration" mean in "As the antenna calibration can be done by a linear function, it has no impact on the following simulation, …"?*

5 Reply: The ideal rectangle backend channel response function is used in the spectrometer simulation.

We are sorry there are some mistakes in the former simulation, a new simulation has been performed and the antenna patterns shown in Fig.8 are used. The section has been rephrased:

"The full-width at half-power points of antenna patterns are used in the following simulation"

10 *P12L15: It is stated that "… the spectra are obtained every 2.5 km." but in Tab. 1 it is shown 1 km. What is the correct value?*

Reply: We are sorry for this mistake. The spectra are obtained every 1 km. It has been corrected.

*P12L15: Is "we use 2.5 km as the vertical resolution" for the retrieval layer? This statement is unclear. Is it the tangent height spacing or antenna FOV vertical resolution or the retrieval layer resolution?*

15 Reply: It is the retrieval layer resolution. The sentence has been rephrased:

"A retrieval grid with 2.5 km spacing is used since it can match the FOV of TALIS well, and cutting down the size of the state vector will give a significant increase in speed (Livesey and Snyder, 2004)"

*P12L7: "The retrieval grid resolution is 1 km below 25 km, 2.5 km below 50 km, and 5 km above 50 km." The choice of 1 km*

20 *for retrieval layer below 25 km should be explained. In my point of view it is not an "optimal" choice for 2 reasons: 1) it is not consistent with the 2.5 km spacing of the tangent heights, and 2) only the 640 GHz radiometer can measure with a resolution better than 1 km.*

Reply: Yes, that is right. We have realized the choice is not an "optimal" one. We have changed the retrieval grid resolution to 2.5 km in new simulation.

*P13L8: I still have trouble with the noise estimation. If I applied Eq 9 to the DBS receiver and taking into account that beta is 2 x spectrometer bandwidth I have: Noise STD at 640 GHz = 3.35 K (=3000/ sqrt(0.1*4.e6)/sqrt(2), with dt = 0.1 sec, beta=4.e6 Hz) This value is smaller than that given by the authors (5 K) by a factor close to sqrt(2). Also the radiometer should be indicated in the sentence: "… to be 2 K, 1.7 K, 1.7 K, and 5 K , respectively." -> "… to be 2 K, 1.7 K, 1.7 K, and*

30 *5 K at 118, 190, 240, 640 GHz, respectively."*

Reply: We are very sorry for misleading you. The fault explanation is discussed above. We have updated the Tsys and recalculated the noise in new simulation. The sentence has been rephrased:

"The expected $1\sigma$ noise is calculated by Eq. (9), the noise is assumed to be 2.2 K, 2.2 K, 2.2 K, and 5.1 K at 118, 190, 240,

643 GHz, respectively"

Reply: Done.

Reply: The error covariance matrix considered here is Sm + Sn (Eq 14 and Eq 15). A sentence has been added:
"The error covariance matrix used in following simulation is the total of $S_n$ and $S_m$."
10   We found the a priori error is set at 110%, it has been corrected in paper.

Reply: Done.

Reply: Yes, the resolution is poorer below 20 km. The figures have been plotted again and a top x-axis is added in order to show the resolution clearly.

Reply: Done.

25   Reply: Done. The sentence has been rephrased:
"However, the 240 GHz product can compensate for the loss of information since the precision is better in in the upper troposphere (error < 1 K for a vertical resolution of 2.5-3 km between 10 and 15 km)"

Reply: Done. In new simulation, the precision at 10 km is also good. A sentence has been added:
"The profile can be retrieved with a single scan precision < 10% from 10 to 55 km and the vertical resolution is 2.5–3 km"

*P21L4-8: I think the diurnal change of ClO should be discussed. The ClO profile corresponds to daytime and the nighttime relative precision will be worse between 30-40 km (ClO vanishes during nighttime). In the polar regions, the relative precision will be high between 20 and 25 km during chlorine activation.*

5    Reply: Done. A sentence has been added:

"Since ClO will vanishes in the middle stratosphere (30-40 km) during nighttime, the precision will be worse in the nighttime. In the polar regions, the relative precision will be high between 20 and 25 km during chlorine activation"

**Reviewer 3**

We would like to sincerely thank the reviewer for his comments. We believe they help us to improve the manuscript significantly and give us many useful suggestions to improve the mission. We have corrected the manuscript according reviewer's comments and answer the reviewer's question point by point below.

15    Reviewer comments are in italic blue, the manuscript modifications are in red. The answer "Done" means that the manuscript has been modified following exactly the reviewer comment.

**Reply to Specific comments**

*Page 1*

20    *Line 8/9: "high precision" means different things to different people (e.g., an airborne in situ instrument would most likely have way higher precision than anything remotely sounded). I'd simply changed it to something like "vertically resolved profile observations"*

Reply: Done.

25    *Line 11: The measurements clearly do not extend to the surface. I'd say "~10-100km, or "the upper troposphere to the mesosphere" or something like that.*

Reply: Done.

*Line 20: "or" -> "and/or"*

30    Reply: Done.

*Line 22: Delete "the"*

Reply: Done.

*Line 23: I suggest "...studies. Satellites can provide dailhy global coverage of the atmosphere. Instruments such as nadir microwave and infrared sounders have been applied..."*

5    Reply: Done.

*Page 2*

*Line 5, add a citation to Barath et al., 1993, doi:10.1029/93jd00798. Also to Waters et al., 1999, doi:10.1175/1520-0469(1999)056<0194:tuaeml>2.0.co;2*

10    Reply: Done.

*Page 3*

*Line 6: Please expand on the "higher" precision. It's not clear to me how the TALIS optical layout specifically enables that. Firstly, to be clear, you do mean "precision" (i.e., random noise-type errors) versus "accuracy" (more "systematic bias" type*

15    *errors), correct? How does the optical layout improve that? Is it because there are fewer beam splitters etc. required? Also, please use terms like "better" and "worse" rather than "higher" and "lower" for terms like precision, accuracy and resolution. This is because when the words are used, having "more" sounds "better". However, when these terms are referred to quantitatively (e.g., with a noise level), then "more" sounds "worse".*

Reply: Thank you for your correction. Yes, precision means random noise-type errors. We have corrected these terms in the

20    paper.

The layout of TALIS can reduce the loss in the front end since there are fewer reflectors and mirrors. It will improve the sensitivity. The accuracy of calibration will also be improved since the targets can cover the four feeds.

*Line 7: Please expand on the 20km displacement. Is this in azimuth (i.e., across the line-of-sight) or in elevetion (i.e.,*

25    *vertically) or some combination of the two? How does this modify your statement that the instrument scans from 0-100km, is it 0-100km for some receivers and 20-120 for others for example?*

Reply: The 240 GHz radiometer has azimuth displacement. The 118 GHz and 190 GHz radiometer have elevation displacement.

For all radiometers, the scan range is 0-100 km, but there will be a sequence. For example, 118 Ghz will scan the surface

30    firstly, then 643 GHz will scan it, 190 GHz is the last. All the data will be recorded as the 118 GHz start to work, so it has no impact on observation.

*Line 10: Please expand on the calibration and instrument design overall. How do the antenna feeds in figure 2 "view the hot target"? Is the box at the end of the arm tipped up to look directly at the target? Does the target fold down to cover the feed horns?*

Reply: The detail has been added to the caption of Figure 1.

5 "The reflector, feeds, and receivers are formed into a whole. The scan driver controls the scan angle. The calibration system is fixed in the satellite. At the beginning, the feeds are covered by calibration targets. Then it will scan the limb. When the system rotates to the top, it will view the cold space"

*Figure 1: Please give more detail on the sketch or in the caption. Which parts of the instrument move to accomplish the scan?*
*Is it just the antenna, just the feedhorn assembly, or both in concert. Alternatively, does the whole spacrcraft nod up and down?*

Reply: Figure 1 has been changed. More details have been added. The scan driver will move with the whole (reflector, feeds, and receivers) to accomplish the scan.

[Figure]

[Figure]

*Table 1: Some of the numbers in the table (and in the discussion that follows) appear contradictory. If the instrument scans from 0-100km in 36 seconds, that's a rate of 2.7km per second. However, the integration time is quoted as "0.1s (1km)". Whatever the case, I've been assuming that all the distances quoted are "vertical", but some may think you're referring to the "along-track" distances (i.e., horizontal). It would be good to be clear on this.*

20 Reply: TALIS spends 3 seconds to scan the hot target, 5 seconds to view the cold space, and 10 seconds to scan from 0-100 km. The total time of 36 seconds defined in paper includes scan and retrace (retrace also records data, it will spend 18 seconds), for one single scan it is 18 second. Yes, it is vertical distances, it has been corrected in paper.

*Page 5*
*Line 6: It is not made clear to me (here or elsewhere) how the "broader bandwidth and finer resolution" is what leads to the improved performance over MLS. I grant that broad bandwidth counts in the lower parts of the atmosphere, but a lot of your improvements come in the mid-stratosphere where the lines are only a 10-100 MHz wide, so bandwidth is not an issue (unless you are bringing in additional lines, which works for ozone, which has many lines, but not water vapor). Are you*

*sure you're capturing the MLS characteristics correctly? As I understand it, the MLS 25-channel spectrometers cover their full bandwidth - as the channels become more widely separated away from the line centers, the channels also get wider increasing the signal to noise. There are no "gaps" in the spectral coverage, which is what the asterisks shown in Figure 3 seem to imply (it might be best to replace the asterisks in Figure 3 with horzontal lines of the correct channel-width, or*

5  *something similar to make that point). Table 2 and Figure 7 of the Waters et al. 2006 reference shows the MLS configuration, please be sure you're capturing it correctly. I suggest you contact the MLS team if you are in any doubt (they are typically fairly responsive). Further, you may have omitted the MLS "Digital Autocorrelator Spectrometers" that I believe provide very fine spectral resolution in the line centers for some species. These contribute to improved performance for MLS in the upper altitudes.*

10  Reply: Thank you for your suggestion. The MLS characteristics are obtained from the reference (Waters et al. 2006). Our simulation configuration follows the information shown in Table 4 and Figure 7 of the Waters et al. 2006. We believe the characteristics are set correctly. The channel-width has been added to Figure 3. In order to compare the performance of spectrometer, all other factors (includes noise) are set the same, only the bandwidth and resolution are different.

The improvements below 20 km mainly come from the bandwidth since the lines are wide. The improvements in the mid-

15  stratosphere and above, the possible reason is the resolution. The 'DAS' spectrometers of MLS are not considered in the paper, it actually improves the performance in the mesosphere. The 25-channel spectrometer has relatively poor resolution.

A statement has been added:

"However, the Digital Autocorrelator Spectrometers of MLS which can improve the performance in the mesosphere are not considered here"

*Pages 6-9, Figures 4-7.*

*It would be really helpful if the colors for each molecule could be consistent from band to band, at least for the main ones that are common to all many plots (O3, H2O, HNO3).*

Reply: Done.

*Page 9*

*Line 13: Poor wording: "Scattering can usually be negelected above the upper troposphere as the atmosphere is largely cloud-free at these altitudes, and such clouds as there are (e.g., Polar Stratospheric Clouds) have particle sizes shorter than the TALIS observation wavelengths.*

30  Reply: Done. The sentence has been rephrased.

*Page 10*

*Line 6: I don't think that's what the definition of "spectroscopy" is. How about "Spectroscopy models and databases allow us*

*to compute the absorption coefficient..."*

Reply: Done. The sentence has been rephrased.

*Line 15: Delete "needs" and change "convert" to "converts"*

Reply: Done.

*Line 16: Change to "... intermediate frequency, folding the upper and lower sideband signals together in consequence".*

Reply: Done.

*Page 11*

*Line 9: I'm not sure about the sqrt(2). Surely it is only needed if the Tsys is defined in terms of single sideband for a double sideband receiver.*

Reply: We are sorry for this fault. The sentence has been deleted. Tsys in Table 1 has been updated in terms of our latest design and a new simulation is performed. Tsys is 1000 K, 1000 K, 1000 K, 2300 K for 118 GHz, 190 GHz, 240 GHz, 643 GHz radiometers. This is a single-sideband value for 118 GHz radiometer, and double-sideband value for other radiometers. The corresponding noise is 2.2 K, 2.2 K, 2.2 K, 5.1 K. The sentence has been rephrased:

"The expected $1\sigma$ noise is calculated by Eq. (9), the noise is assumed to be 2.2 K, 2.2 K, 2.2 K, and 5.1 K at 118, 190, 240, 643 GHz, respectively"

*Line 30, Equation 14. Which of the covariance matrices are shown summarized as "Error" in the figures that follow? Looking at the plots I actually think it's the total of Eq 14 and Eq 15, but I'm not sure.*

Reply: Yes, the error is the total of Eq 14 and Eq 15. A sentence has been added:

"The error covariance matrix used in following simulation is the total of $\boldsymbol{S}_n$ and $\boldsymbol{S}_m$"

*Page 12*

*Line 10: I'm worried about the authors' choice to ignore the finite size of the field of view in the discussion that follows. It really is no additional hardship with ARTS to add that step, and would make for a far more meaningful study. If nothing else, I urge them to quantify the impact it has, which will not be on the precision so much as the vertical resolution.*

Reply: We have added the antenna patterns shown in Fig.8 to our new simulation. The sentence has been rephrased:

"The full-width at half-power points of antenna patterns are used in the following simulation"

*Line 14: Are they refering to the vertical resolution of the radiance obsevations or of the state vector vertical grid here? In either case, why is the FOV for TALIS or MLS germain to that choice? The tradeoff they are referring to with the Livesey*

*and Snyder paper is not really related to the resolution of the reporting grid, rather to the resolution of the information yield (e.g., as represented by the width of the averaging kernels). Also, if the spectra are obtained avery 2.5km what is the integration time assumed, and the total time taken for the vertical scan. Again, as with the numbers in Table 1, there is an inconsistency here. Fundamentally there are multiple "resolutions" at play here: The spacing of the retrieval grid, the*

5  *spacing of the radiances, and the degree of information content in the Level 2 data products. Please try to be clear which you are referring to in each case.*

Reply: We are sorry there are some mistakes in the former simulation, and have been corrected in new simulation. The spectra are obtained every 1 km, it is according to the numbers in Table 1. The spacing of the retrieval grid is 2.5 km in order to match the field of view. The sentence has been rephrased:

10  "A retrieval grid with 2.5 km spacing is used since it can match the FOV of TALIS well, and cutting down the size of the state vector will give a significant increase in speed (Livesey and Snyder, 2004)"

*Paragraph starting at line 14, and through to end of 4.1: I think more information is needed in this section. Firstly, from where did the various atmospheric profiles assumed as "truth" and "apriori" come from? It just says "mid-latitude summer*

15  *conditions". Are MLS data the source of the profiles? If so, please be explicit about the version of MLS data used, latitude/date range used, application of quality screening, assmptions made to go from pressure to altitude as the vertical coordiante etc. What about the species that MLS doesn't measure, ro doesn't measure well? If the profiles instead come from a model, again give all the information needed to suppor reproducibility. Secondly, in the retrieval calcaultions that follow, are the various molecules retrieved simultaneously from all the bands together, or are some subset (which) of the molecules*

20  *retrieved independently from each band? Are there also spectrally flat (or simple linear or quadratic form) "Baseline" or "Extinction" terms included in the retrievals. These are needed to account for instrumental/forward model issues, and can impact the precision/resolution trade.*

Reply: The profiles are provided by ARTS which is extracted from FASCOD. The BrO and $HO_2$ profiles are not included in ARTS-XML-DATA, so we use MLS L3 monthly averaged data (observed from 20°N-30°N in July, 2018). The molecules

25  are retrieved simultaneously from each band. Spectrally flat is not included. The sentence has been rephrased:

"A mid-latitude summer atmospheric condition extracted from FASCOD which is provided by ARTS (profiles of BrO and HO2 are from MLS L3 monthly averaged data, 20°N-30°N, July, 2018) is chosen to perform the simulation"

"The molecules are retrieved simultaneously from each band"

30  *Page 13*

*Line 3: Again, where does the "typical profile" come from*

Reply: Answered above.

*Line 4: In my experience the Gaussian form is a poor choice for the off-diagnoal terms, as it gives a normal equation matrix that is close to singularity. Instead an exponential form is generally preferred (and corresponds to a 1st-order Tikhanov smoothing).*

Reply: Done, in our new simulation, the exponential form is used.

*Line 10: How much averaging is the factor of 10 equivalent to? Presumably 100x as many measurements, correct? Please put that in terms a reader will readily understand (e.g., "equivalent to a 10-degree latitude weekly zonal mean" or whatever it is).*

Reply: Done. A statement is added:

10 "equivalent to a 10-degree latitude weekly zonal mean"

*Line 16: As stated above, I find it hard to believe the spectral resolution and bandwidth alone accounts for the MLS/TELIS differences shown, especially above ~20-30km. Are the authors sure they have correctly captured the MLS spectral coverage (see above)? The effective bandwidth (i.e., the width of the spectral regions that actually contain information) are essentially*

15 *the same between the two instruments. Are the authors using the same system temperatures for both instruments (as strongly implied by line 16) or are they in fact changing those numbers too, if so that's the likely cause. Note the discussion of sqrt(2) above, I'm pretty sure the MLS team quotes thir system temperatures without invoking that factor. This is an important issue that warrants greater investigation.*

Reply: Yes, the MLS spectral coverage is captured from (Waters et al. 2006). In simulation, only resolution and bandwidth

20 of the two instruments are different, and the noise assumed are the same.

*Figure 9 (and others): Please make it clear what "error" means in this context. Is it the sqrt-diagonal of one of the covariance matrices quoted above (or the sum of several)? Please be clear. I suggest the authors add some clear metric of vertical resolution to the right-hand figure (e.g., full width at half maximum, or degrees of freedom per km), with a separate*

25 *top x-axis.*

Reply: The error covariance matrix has been stated:

"The error covariance matrix used in following simulation is the total of $S_n$ and $S_m$"

Done. The figures have been plotted with the top x-axis to show the FWHM.

30 *Page 15*

*Line 1: Again, which covariance matrix!*

Reply: Answered above.

*Figure 12 and its siblings: I really don't see the point of showing a single retrieved noisy profile here, it's but one representative retrieval exhibiting noise and risks giving the uneducated reader the impression of some kind of systematic biases that cannot be reduced. Related to that, is the "Error" term again the sqrt of some covariance diagonal, or is it the difference between retrieved and true (I don't think so, looking at the results, but some readers may get confused). I'd just simply show the retrieved precision, and the typical profile (combining the first two panels into one with only two lines).*

Reply: Done.

*Page 16*

*Line 8: "less" -> "poorer"*

Reply: Done.

*Page 24*

*Line 5: As discussed, it's essentially impossible for the reader to "see" the 3km resolution from your figures. Please add some suitable line/plot to make that clrer.*

Reply: Done.

*Line 10: "Significantly average" -> "Significant averaging"*

Reply: Done.

*Page 29*

*Line 5: Change first sentence to: "Seven species show high sensitivity, sufficient for scientifically useful single profile retrievals".*

Reply: Done.

*Line 23/24: Just a note, the Zeeman effect will indeed be critical to the correct retrieval of mesospheric temperature, though it will probably not impact the precision/resolution results shown here.*

Reply: Thank you for your note, we will add Zeeman effect in future study.

[revised manuscript text omitted]